# A Pixel-Wise k-Immediate Neighbour-Based Image Analysis Approach for Identifying Rock Pores and Fractures from Grayscale Image Samples

Pradeep S. Naulia [1], Arunava Roy [2], Junzo Watada [3,*] and Izzatdin B. A. Aziz [1]

1 Center for Research in Data Science (CeRDAS), Universiti Technologi Petronas (UTP), Seri Iskandar 32610, Malaysia
2 Department of Computer Science, The University of Memphis, Memphis, TN 38119, USA
3 Information, Production and Systems Research Center, Waseda University, Wakamatsu, Kitakyushu 8080135, Japan
* Correspondence: watada@waseda.jp; Tel.: +81-90-3464-4929

**Abstract:** The purpose of the current study is to propose a novel meta-heuristic image analysis approach using multi-objective optimization, named 'Pixel-wise k-Immediate Neighbors' to identify pores and fractures (both natural and induced, even in the micro-level) in the wells of a hydrocarbon reservoir, which presents better identification accuracy in the presence of the grayscale sample rock images. Pores and fractures imaging is currently being used extensively to predict the amount of petroleum under adequate trap conditions in the oil and gas industry. These properties have tremendous applications in contaminant transport, radioactive waste storage in the bedrock, and $CO_2$ storage. A few strategies to automatically identify the pores and fractures from the images can be found in the contemporary literature. Several researchers employed classification technique using support vector machines (SVMs), whereas a few of them adopted deep learning systems. However, in these cases, the reported accuracy was not satisfactory in the presence of grayscale, low quality (poor resolution and chrominance), and irregular geometric-shaped images. The classification accuracy of the proposed multi-objective method outperformed the most influential contemporary approaches using deep learning systems, although with a few restrictions, which have been articulated later in the current work.

**Keywords:** deep CNN; clustering; image processing; micro-pores; micro-fractures; pixel intensity; porosity





## 1. Introduction

Carbonate rock reservoirs are believed to contain nearly 50% of the world's hydrocarbons. The carbonate rocks are dominated by a few carbonate minerals (mainly calcite and dolomite) and contain additional traces of minerals such as silica, detrital grains, phosphate and glauconite. On the other hand, carbonate rocks may contain organic residues and some cementing material. Petrography helps identify grains, leading to the detailed classification of rocks, the determination of the types of deposition and the indicators of historical post-depositional alteration (diagenesis). It also helps determine the timing of porosity development. Comparing these grains with photographs allows us to readily identify the essential properties/features that can potentially predict the amount of petroleum under adequate trap conditions in the oil and gas industry. In this study, the pore/fracture space component using multi-objective classification is based on microscopic studies, particularly of Malaysian carbonate rocks. Here we focus on the image analysis of rock image samples to automatically reach a conclusion on pores or fractures contained in the image sample instead of illustrating the quantitative petrophysical method. Hence, we will not work on overall petrography analysis but only its segment related to image analysis.

Porosity is widely considered a fundamental parameter for reservoir characterisation and is used in many hydrogeological and geophysical calculations. For crystalline rock, it can be calculated as the sum of micro-fractures, irregular micro-pores and fluid inclusions [1]. Porosity also determines the storage capacity of a specific reservoir. To our knowledge, most hydrocarbon reservoirs belong to sedimentary rock formations in which porosity values vary between 10% and 40% for sandstones and 5% and 25% for carbonates [2]. The micro-pores are spherical or nearly spherical [3]. In contrast, fractures are assumed to maintain near linearity (i.e. their length is longer than their width). Total porosity can be divided into two types: connected (characterised using flow and diffusion) and unconnected (because of the presence of isolated pores) [1]. Unconnected spherical or near-spherical pores along with micro-porosity can effectively change the elastic properties of the rock frame, making porosity an important parameter for reservoir characterisation in terms of estimating elastic moduli [3].

The current study aims to propose a novel meta-heuristic image analysis approach using the multi-objective algorithm approach to alleviate the previous difficulty in identifying pores and fractures (natural and induced, even at the micro-level) in the wells of a hydrocarbon reservoir. Furthermore, we present better identification accuracy in the grayscale sample rock images in this work.

The high-level idea of the proposed methodology is multi-objective, i.e. the primary objective is to find the neighbouring dark pixels of every discovered dark pixel within a specific region and create a bag of contiguous pixels. A secondary objective is to classify the shapes obtained from the pixel bag into pores or fractures based on a certain chosen hyper-parameter. This methodology was compared with the CNN (convolution neural network)-based [4,5] approach, which is considered a state-of-the-art framework for image classification. CNNs comprise pixel-to-pixel and end-to-end architectures, which use a large image dataset with features and tuneable neural network parameters. Our proposed method can obtain more accurate results than a deep learning CNN, which thus far has been considered the most efficient method for identifying pores and fractures in hydrocarbon reservoirs.

## 1.1. History of Existing Research Work

Researchers have undertaken these challenges and have tried to deploy various intelligent techniques, such as deep learning approaches. In an early study, Lucia et al. [6] developed a carbonate rock porosity detection method that uses the visual inspection of a sample rock as an input. This method [6] relies on the expertise of a scientist for porosity detection. In a ground-breaking work, Ehrlich et al. [7] also developed an image analysis technique that considers colour images (RGB) to detect micro-pores and fractures, although we are unconvinced of its efficacy in the presence of grayscale images. Similarly, the work of Funk et al. [8] describes sample rock pore size distributions by analysing the petrophysical properties derived from a rock's colour images. Van den Berg et al. [9] showcased an image-processing method for porosity detection that uses high-quality image inputs. The pore and fracture sizes are quite large and visually identifiable [9]. In another study of semi-automated rock texture identification image-processing techniques, Perring et al. [10] considered high-resolution images as input samples. We believe that this method might not be able to replicate its performance in the presence of low-quality grayscale images. Additionally, several contemporary studies [11–14] on pore/fracture detection used simple image analysis techniques. In another study, Chen, J., et al. [15] developed an automated image processing-based method to identify rock fracture segmentation and its trace quantification using a CNN-based model to extract the skeleton of the cracks and a chain code-based method to quantify the fracture traces. They extended this research [16] into rock trace identification using a hybrid of the synthetic minority oversampling technique, random search hyper-parameter optimization and gradient boosting trees. Chen, J., et al. [16] proceeded to propose a novel CNN-based water inflow evaluation method that emulates a typical field engineer's inspection process. This method classifies the undamaged and damaged regions and segments the detailed water inflow damage to

the rock tunnel faces. Most of the abovementioned studies used higher-resolution images; however, we suspect that the above proposed methods may not be able to replicate their performances in the presence of lower-quality grayscale images.

In the contemporary literature, a few studies employing machine learning, such as the support vector machine (SVM) classifier and deep learning, achieved the pore/fracture detection of various reservoir rocks, and the noteworthy works are illustrated below. Leal et al. [17] used the fractal dimensions of images, gamma rays and resistivity logs as the input dataset to an SVM classifier to detect micro-fractures. However, the detection accuracy was not the best in class, and we suspect that it might suffer from poor image quality with grayscale. Abedini et al. [18] employed deep learning and autoencoder techniques to identify the pores/fractures from rock images. Although they used visually discriminating images [18], their detection does not require advanced techniques such as deep learning. This strategy may fail in the presence of a tiny training image sample set with high similarities among images (low divergence). From the above study, it is evident that the existing contemporary detection and classification methods need further advancements to improve their accuracy, particularly in the presence of a tiny training set. A tiny training set is a real problem challenging the accuracy of deep learning strategies for low-quality image samples. Hence, in the current work, we tried to devise a method that works well with a small number of low-quality grayscale image samples.

The current work addresses the identification of pores and fractures that do not have a fixed geometric shape; hence, before discussing the principal findings of the present work, a brief discussion of the recent noteworthy findings of machine learning regarding the identification of irregularly shaped objects would be useful. In an early work, Viola and Jones [19] presented a face detection framework capable of detecting faces from an image and hence, suitable for detecting objects with near-fixed geometric structures. This framework mainly focuses on collecting Haar-like features along with the histograms of oriented gradients (HOGs) of the objects with fixed geometric shapes for their possible detection. However, these Haar and HOG features were found to be relatively inefficient in detecting irregularly shaped objects, which is highly required in the current context as the shapes of pores and fractures are highly irregular. Lienhart and Mayst [20] introduced limited and arbitrary rotations for object identification, which are relatively inefficient in identifying objects of irregular shapes. David Gerónimo et al. [21] proposed a pedestrian classifier based on Haar wavelets () and edge orientation histograms (EOHs) with AdaBoost; which is compared to SVMs using HOGs. The results show that the HW + EOH classifier achieves comparable accuracy but at a much lower processing time than the HOG. In another contemporary work, Smith et al. [22] introduced a ray feature set that considers image characteristics at distant contour points to gather information about an object's shape. However, this strategy is often computationally expensive and may often be challenging for identifying the necessary context-specific ray features [22].

Modern deep learning and other machine learning classifiers require many training samples to correctly extract shape-related information necessary for identifying a specific object. However, the scenario becomes more complex in the presence of a low-quality, tiny training image set of extremely deformable objects (such as pores and fractures in the current case) since identifying the shape-related features necessary for their unique identification often becomes extremely difficult. Hence, a low-level pixel intensity-based image analysis approach could be more computationally efficient and accurate if used to identify irregular objects instead of the deep learning and machine learning counterparts.

*1.2. Contribution of the Current Work*

Visual quality evaluation has traditionally focused on the perception of quality from a human subjects' viewpoint, which motivates us to characterise the effect of image quality on current computer vision systems. However, these notions of image quality may not be directly comparable, as a computer can often be fooled by images that are perceived by humans to be identical [23], or in some cases, the computer can recognise images that

are indistinguishable from noise by a human observer [24]. Hence, how image quality often affects the accuracy of a computer vision technique must be separately considered. In this study, we found that the accuracy of the existing methodologies [18,25] substantially deteriorates with grayscale images. Hence, an approach that can work successfully given a tiny set of low-quality image samples as input is urgently needed. Here, we propose an approach that delivers robust results even if the quality of the input image is severely deteriorated (low quality and distorted). We addressed this approach in the following sections of the current article.

To address this issue, we analyzed the brightness of every pixel and classified it as bright or dark based on some preconceived notions, which are flexible depending on the respective problem definitions, image sample quality, and necessary geophysical properties of the rock sample used. Next, the pores and fractures are identified from the input grayscale images based on some shape-related preconceived logic (pores spherical or near-spherical, while the fractures are more linear in shapes, in general, as per the contemporary literature) and the pixel classification results.

The proposed approach has been tested with CT-scan images of carbonate rocks, and results are found more accurate than those of the deep-CNN and other contemporary approaches, as is shown later in the present work.

### 1.3. Paper Structure

The rest of the paper is arranged as follows: Section 2 describes the methodology for micro-pore and fracture identification from the input grayscale CT-scan image samples; Section 3 shows the implementation of the proposed method and presents detailed comparisons with the noteworthy contemporary approaches from the accuracy point of view. The main points of the present study are articulated in Section 4.

### 2. Materials and Methods—Porosity Classification and Fracture Identification

This section elaborates on the developed image analysis methodology for porosity classification and fracture identification of rock. To make our proposition more robust, we assume the input images are of low quality, primarily grayscale, and less statistically divergent. Here, less statistical divergence in the dataset ($\mathcal{M}$) points to the high similarity among its various image samples, which subsequently might make the mini-batch samples very similar and might lead to the mode collapse for the deep learning models.

The basic idea of the proposed methodology is multi-objective—to find the $k$-immediate neighboring dark pixels of every discovered dark pixel within a specific region in an input grayscale image $G_n \in M$ and then to classify it as a pore or a fracture. This is because, from the image processing point of view, we consider a pore or a fracture as a region within an image that contains a large number of contiguous dark pixels, even if not all of them are dark. Hence, we name the developed algorithm as "Pixel-wise $k$-Immediate Neighbors" (Pixel-wise $k$-IN) approach, where the set of $k$-immediate neighboring pixels ($k$-IN) of any specific pixel located at $(x, y)$ in $\mathcal{G}_n$, $n = 0, 1 \cdots$ are listed as $k$-IN$(x, y)$ = {$(x − 1, y − 1)$, $(x, y − 1)$, $(x + 1, y − 1)$, $(x −1, y)$, $(x + 1, y)$, $(x − 1, y + 1)$, $(x, y + 1)$, $(x + 1, y + 1)$}, where $x, y$, $x − 1, x + 1, y − 1, y + 1 \in \mathbb{Z}^+$. In short, the $k$-IN of $(x, y)$ can be defined as $k$-IN$(x, y)$ = {$(x \pm i$, $y \pm j)$; $\forall x, y \in \mathbb{Z}^+$; $i, j = 0, 1$; $i, j \neq 0$ simultaneously}.

For more clarity, in Figure 1a, the set of $k$-INs computed for pixel 6 is {1, 2, 3, 5, 7, 9, 10, 11}, where that for the pixel 1 is {2, 5, 6}. The procedure to classify every individual pixel in an input grayscale image as dark or bright is shown in Section 2.1. Later, we articulate the procedure to identify the $k$-immediate neighboring dark pixels of any specific pixel and classify a region containing a large number of contiguous dark pixels as pore or fracture, as is shown in Section 2.4. and Algorithm 2.

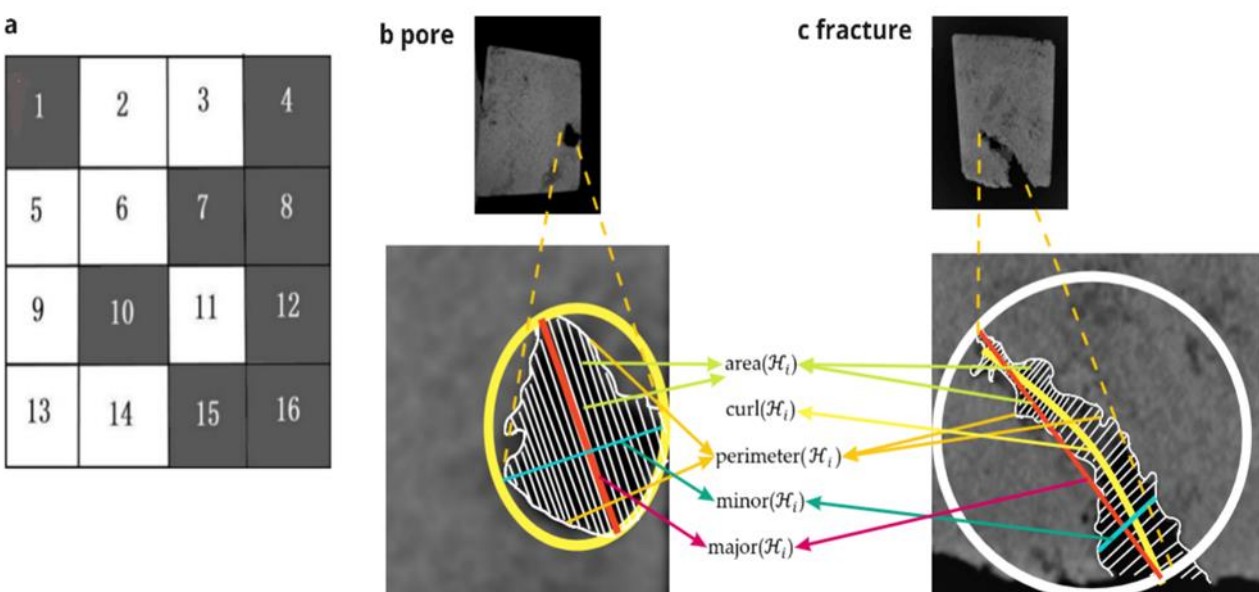

**Figure 1.** (**a**). Step-wise procedure to find *k*-INs of any pixels. **area**($\mathcal{H}_i$), **major**($\mathcal{H}_i$), **minor**($\mathcal{H}_i$), **perimeter**($\mathcal{H}_i$) and **curl**($\mathcal{H}_i$) of a fracture and pore are shown in (**b**,**c**), respectively. Quite evidently, the $\frac{\text{major}(\mathcal{H}_i)}{\text{minor}(\mathcal{H}_i)}$ value of a fracture is larger than that of a pore.

### 2.1. Classifying Bright and Dark Pixels

All the images are in grayscale with an average pixel intensity of 101/image. Each pixel in a grayscale image represents only the amount of light or the intensity information. Grayscale images are a black-and-white or gray monochrome that is composed exclusively of shades of gray. We measured the intensity ($\mathcal{L}_{i,j}$), the measure of brightness/darkness) of any pixel $\mathcal{P}_{i,j} \in \mathcal{G}_n \in \mathcal{M}$. We classify a pixel as dark or bright if $\mathcal{L}_{i,j} \in [0, I]$ and $\mathcal{L}_{i,j} \in (I, 255]$ respectively, where *I* will be the threshold with range $0 < I < 255$ (see Algorithm 1). Here, $\mathcal{L}_{i,j} = 0$ means fully dark, whereas $\mathcal{L}_{i,j} = 255$ classifies as fully bright. *I* mostly depends on the resolution of the available image dataset, and we humbly encourage the readers to choose suitable *I* values based on their expert judgments. In easy words, we classify pixels (as bright or dark) based on their computed $\mathcal{L}_{i,j}$ values. The proposed image processing technique will be applied only in the effective region of every individual image, which we will term as the red boundary. The proposed image processing technique understands the red boundary by analyzing the brightness of the bordering pixels based on their $\mathcal{L}_{i,j}$ values.

---

**Algorithm 1** Pixel Classification Procedure

---

**Input:** Grayscale image $\mathcal{G}_n$ of rock contain $l \times s$ pixels.
**Variables** & **Parameters:** $\mathcal{L}_{i,j}$, $\mathcal{P}_{i,j}$ $\forall\, i \le l, j \le s$;
**Outputs**: Classification of every pixel as dark and bright.
1: **procedure** PIXEL_CLASSIFICATION($\mathcal{G}_n$)
2:    **for** (*i* = 0; *i* < *l*; *i*++) **do** {
3:        **for** (*j* = 0; *j* < *s*; *j*++) **do** {     ▷ starting pixel at *i*,*j*-=0, 0 (top left)
4:            **if** ($\mathcal{L}_{i,j} \in [0, I]$) **then** $\mathcal{P}_{i,j} \in \mathcal{G}_n$ is dark
5:            **else if** ($\mathcal{L}_{i,j} \in (I, 255]$) **then** $\mathcal{P}_{i,j} \in \mathcal{G}_n$ is bright
        }
    }
6:    **End**.

---

### 2.2. Calculating Similarity among the Images

More often than not, it is quintessential to measure the similarity/dissimilarity between different image samples within the dataset under consideration to scrutinize and

understand the reasons behind every specific unusual or unsatisfactory outcome. The resulting accuracy of various image analysis methods (including deep learning) is often found relying on their adopted learning techniques and might suffer because of the similarity/dissimilarity between the individual image samples in the training and testing sets. For example, a machine vision system fails to recognize a set of cars if it is trained to identify a set of a few specific animals. Apart from that, a reasonable similarity measure between image samples is often considered as one of the essential parts of most of the image classification systems [26], and since, in the current work, our prime objective is to identify the pores and fractures within the grayscale rock images, we have presented our similarity measure as below.

The present subsection describes the adopted similarity evaluation technique in detail. We assume $\mathcal{M} = \{\mathcal{G}_n\}$; $1 \leq i \leq N$ is the image dataset, and each image contains $m \times n$ pixels. Each image of $\mathcal{M}$ is represented by the pixel intensity vector $\mathcal{L}^i = \{\mathcal{L}^i_{p,q}\} \subset \mathbb{R}^{mxn}$; $1 \leq p \leq m$; $1 \leq q \leq n$, where $\mathcal{L}^i_{p,q} \in \mathbb{R}$ is the intensity value (scalar) of the pixel $(p, q)$ of an image $\mathcal{G}_i$, which is evaluated as in Section 2.1, Algorithm 1.

Next, our job is to find the $k$ most similar images to a given image $\mathcal{G}_i$ based on some defined similarity criterion. We further assume that all images have equal prior probability, and the query image ($\mathcal{G}_i$) is represented by the pixel intensity vector $x_i$. From the probabilistic point of view, each pixel intensity vector contains $m \times n$ realizations of *i.i.d* (independent and identically distributed) random variables $\mathcal{L}^i_1; \ldots; \mathcal{L}^i_{mn}$, which follow a parametric distribution with probability density function $p(\mathcal{L}|\theta)$; $\theta \in \mathbb{R}^d$. Given that $\hat{\theta}$ is a consistent estimator of the parameter vector $\theta$, it is quite evident that for an image $\mathcal{G}_j$ to be the most similar to $\mathcal{G}_i$, whose parameter vector $\theta_j$ leads to the maximization of the following log-likelihood function, then:

$$j = \underset{r}{\operatorname{argmax}} \frac{1}{mn} \sum_{p=1}^{m} \sum_{q=1}^{n} p\left(\mathcal{L}^r_{pq} \Big| \theta_r\right) \tag{1}$$

By applying the weak law of large numbers to the Equation (1) with $m, n \to \infty$, we obtain the following:

$$j = \underset{r}{\operatorname{argmax}} \, \mathbb{E}_{p(\mathcal{L}|\theta_i)} \log p(\mathcal{L}|\theta_r) \tag{2}$$

$$= \underset{r}{\operatorname{argmax}} \int_L p(\mathcal{L}|\theta_i) \log p(\mathcal{L}|\theta_l) d\mathcal{L} \tag{3}$$

where $\mathbb{E}_{p(\mathrm{L}|\theta_i)}(\cdot)$ is the expectation with respect to $p(\mathrm{L}|\theta_l)$ and $L$ is the domain of $p(\mathrm{L}|\cdot)$. By observing $p(\mathrm{L}|\theta_i)$ as an independent term for the maximization, Equation (3) can be rewritten as follows:

$$j = \underset{r}{\operatorname{argmin}} \left\{ -\int_L p(\mathcal{L}|\theta_i) \log p(\mathcal{L}|\theta_r) d\mathcal{L} \right\} \tag{4}$$

$$= \underset{r}{\operatorname{argmin}} \int_L p(\mathcal{L}|\theta_i) log \frac{p(\mathcal{L}|\theta_i)}{p(\mathcal{L}|\theta_r)} d\mathcal{L} \tag{5}$$

In Equation (5) $\mathcal{D}_{\mathrm{KL}}(.||.)$ is between $p(\mathcal{L}|\theta_i)$ and $p(\mathcal{L}|\theta_r)$ or $\mathrm{D}_{\mathrm{KL}}(p(\mathcal{L}|\theta_i)||p(\mathcal{L}|\theta_r))$. Hence, in the asymptotic case, maximum likelihood selection is equivalent to the minimization of the $\mathcal{D}_{\mathrm{KL}}(.||.)$ and subsequently, the image similarity can be calculated using $\mathrm{D}_{\mathrm{KL}}(p(\mathcal{L}|\theta_i)||p(\mathcal{L}|\theta_r))$.

### 2.3. Dataset Specification and Pre-Processing

In this study, we focus on image analysis of the rock images samples to automatically arrive at a conclusion on the type of pores or fractures instead of the quantitative petrophysical method. Thus, we will not work on overall petrography analysis but only its segment related to image analysis.

The pore/fracture space component classification is based on a microscopic study from peninsular Malaysia in Kinta Valley, Perak. The pore types include mouldic, intraparticle,

interparticle, fractured and vuggy porosity. The initial observation indicated that the pore system is isolated in nature. From the perspective of visual image analysis, in most cases, the shape and size of the micro-pores and fractures were highly irregular in shape. In contrast, our techniques showed fractures to be near linear (i.e., length is bigger than width). They were also found to be well cemented. Fractures were found to be enlarged by solution activity or can be healed by secondary calcite or sparite. Based on the previous experience, in general, we found the lengths of the fractures are much greater than their widths. The shape variations of a fracture from that of the circles and ellipses of same perimeter are found much greater than that of the pores. On the contrary, as expected the shapes of the pores were found to be spherical or near-spherical, and in general, their sizes are relatively smaller than those of the fractures but still irregular in shape from the computer vision context. Further, the image data generated in the current work shows limited divergence among most of the samples, i.e., the visual quality of most of the samples is found to be very similar. Low divergence samples have been identified as a major problem for the existing deep learning methodologies in the presence of a small number of grayscale image samples with low divergence, especially in the case of the current image samples.

Optical thin section and Scanning Electron Microscopy (SEM) images were used to characterize and describe the different components and structures within the carbonate rock. Before diving into details of the proposed algorithm, we would like to discuss the specification of the rock image samples used in the current study to understand its utility more clearly. We used the CT-scan images of a carbonate rock slab of length 256 mm, breadth 1 mm, and width 1 mm taken at every 1 mm height in all $xy$, $yz$ and $zx$ planes (see Figure 2). We call every image a micro-image. Hence, the image data set contains a total of $256 \times 3$ number of micro-images of varying numbers of pixels, which are then resized to fixed $1280 \times 1064$ square pixels, where each pixel area is approximately limited to 4.4 $\mu m^2$, and one micro-image creates a field view of $1 \times 1$ mm. The spectral noise of the collected CT-scan image samples was reduced using a median filter, and the contrast between features was enhanced with a linear contrast stretch. Most of the individual sample micro-images look very similar (see Figure 2). This observed high visual similarity among various sample micro-images is due to the small size of the sample carbonate rock slab (256 mm) considered during CT-scan imaging. High visual variations among the sample micro-images can be experienced if the chosen size of the rock sample is larger (at least around a couple of meters). Further qualitative analysis of each 3 arbitrary planes ($xy$, $yz$ and $zx$ ) shows that some of the individual pores are large and isolated.

All the images are in grayscale with an average pixel intensity of 101/image. Each pixel in a grayscale image represents only the amount of light or the intensity information. Grayscale images are a kind of black-and-white or gray monochrome that is composed exclusively of shades of gray. The subsequent values are respectively calculated as 50, 120 and 255 (see Section 2.1). Based on the computed values, we classify every individual pixel using Algorithm 1.

Each image has contiguous dark regions lying on its borders, which are neither pores nor fractures (see Figure 2), and we must exclude them before beginning the computations. Hence, to eliminate these dark regions during computation, we manually marked their borders with red ($R = 255$, $G = 0$, $B = 0$) (see Figure 2) during data pre-processing, and we call the regions enclosed within the red boundaries as the "effective regions." The proposed image processing technique understands the red boundary by analyzing the brightness of the bordering pixels based on their $\mathcal{L}$-values (see Section 2.1). We have kept data distribution (size of training, validation, test splits) as 60:20:20 for deep-CNN. For our proposed method we did not follow any data distribution since the method does not call for any training. Hence, to compare both we have to translate all deep-CNN metrics into Accuracy (Accuracy $= \frac{(TP+TN)}{(P+N)}$) so as to match it to results captured from our proposed methods which are displayed in the Table 1.

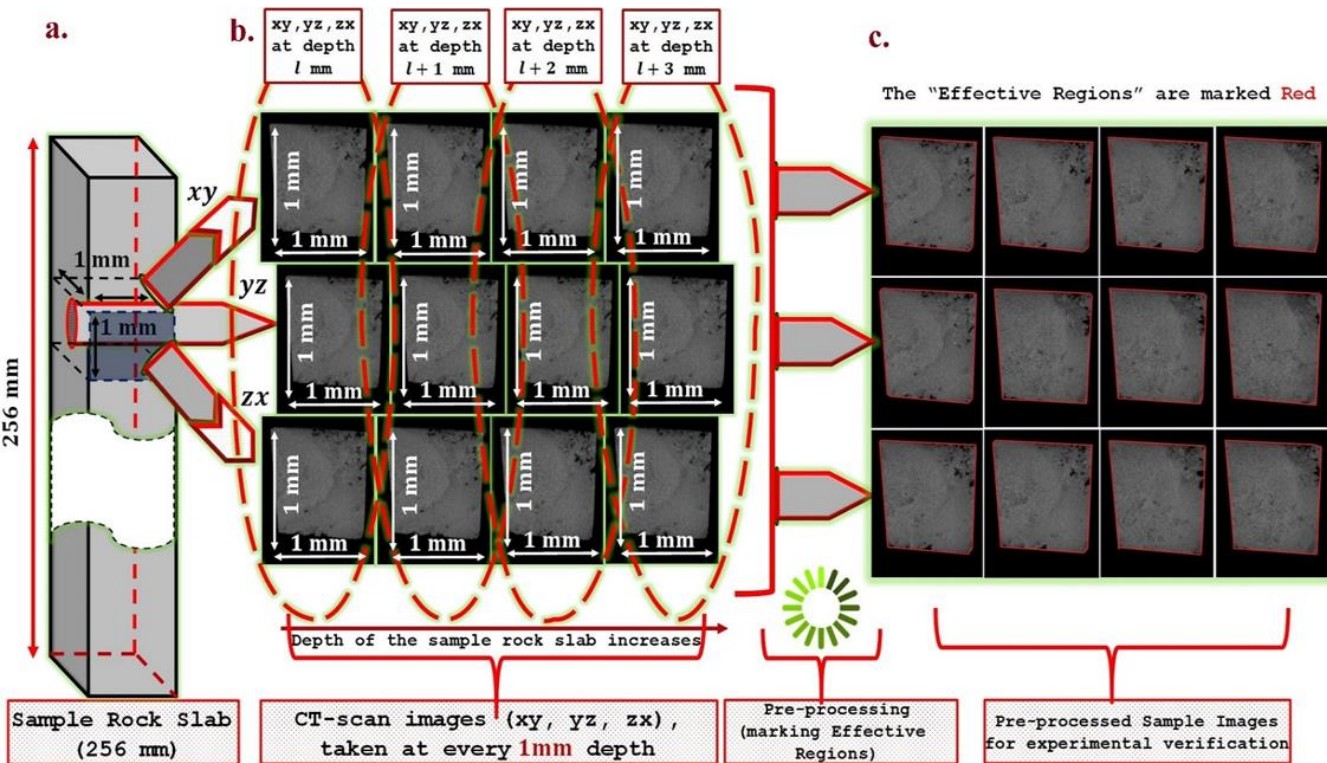

**Figure 2.** (**a**) Collection of raw CT-scan image samples of carbonate rock. The samples are grayscale images. (**b**) Sample raw CT-scan images. (**c**) Pre-processed CT-scan image samples with already marked boundary with red color.

**Table 1.** Comparing the accuracy of the proposed $k$-IN with that of the supervised deep-CNN approach. Here, we found the accuracy of the proposed approach outperforms the deep-CNN.

|  | Accuracy | Error Rate |
|---|---|---|
| Deep-CNN (Masked RCNN) | 0.093 (mAP) | -NA- |
| Deep-CNN (Masked RCNN Detectron2) | 0.59 (mAP) | -NA- |
| Deep-CNN (Yolo5 Conf 0.5–0.9) | 0.395 (mAP) | -NA- |
| Custom Object CNN | 24.9% | 85.1% |
| $\alpha = 1, \beta = 1.11, \gamma = 0, \delta = 0$ | | |
| $k$-IN | 84.9% | 15.1% |
| $\alpha = 1.03, \beta = 0.95, \gamma = 0.43, \delta = 0.43$ | | |
| $k$-IN | 89.1% | 10.9% |
| $\alpha = 1, \beta = 1.11, \gamma = 0.45, \delta = 0.45$ | | |
| $k$-IN | 81.2% | 18.8% |
| $\alpha = 0.9, \beta = 1, \gamma = 0.45, \delta = 0.45$ | | |
| $k$-IN | 79.9% | 21.1% |

## 2.4. Pixel-Wise k-IN Approach to Determine Rock Fractures and Pores

As we discussed earlier, our primary objective is to find the regions ($\mathcal{H}_i; \forall_i$) (with irregular shapes) within a given grayscale image ($\mathcal{G}_i \in \mathcal{M}$) that contains all contiguous dark pixels. Our secondary objective it is to later classify the region as a pore or a fracture. In the current work, we identify a region as $\mathcal{H}_i$ containing a contiguous collection of 100 or more dark pixels (at least of area $100 \times 4.4 \ \mu m^2$, see Figure 1a,c). However, readers are encouraged to modify this convention based on the quality of available rock image

samples, problem definitions, and necessary geophysical properties of the respective rock sample (such as porosity, etc.).

- Information about the shape geometry of $\mathcal{H}_i$ is extracted from the significant shape descriptors such as the shape boundary (sensing the abrupt change in the intensity level using intensity gradients), perimeter($\mathcal{H}_i$), major($\mathcal{H}_i$), minor($\mathcal{H}_i$), shape variations of $\mathcal{H}_i$ with respect to circle ($\mathcal{V}_c(\mathcal{H}_i)$) and ellipse ($\mathcal{V}_\mathcal{E}(\mathcal{H}_i)$) of the same perimeters, etc. The induced logic to classify any $\mathcal{H}_i$ as a pore or a fracture is discussed later in the current subsection.

### 2.4.1. Fracture Shape Geometry in a CT-Scan Image Sample

A fracture is defined as any separation in a geological formation that divides the rock into two or more pieces [27], which in the subsequent grayscale image can be seen as a region with its length much larger than its width and containing a large number of contiguous dark pixels. From a reservoir point of view, fractures are planar discontinuities or deep fissures in rocks due to mechanical deformations or physical diagenesis [17,28]. Fractures are often found enlarged by the solution activity or can be healed by secondary calcite or sparite. Based on the previous experience, in general, we found the lengths of the fractures are much larger than their widths, and hence, in the current study, we assume that, to be a fracture, $\mathcal{H}_i$ should follow $\frac{\text{major}(\mathcal{H}_i)}{\text{minor}(\mathcal{H}_i)} > \beta$ (=1.3 in the current study) and in general, their shapes are much deviated from a circle or ellipse of the same perimeter as $\mathcal{H}_i$. In the current work, we measure the length of $\mathcal{H}_i$ as major($\mathcal{H}_i$), whereas the width of $\mathcal{H}_i$ is measured as minor($\mathcal{H}_i$) (defined later). The shape variations of a fracture from that of the circles and ellipses of same perimeter are found much larger than for pores. In the current work, we encourage the readers to decide the $\beta$ value based on the porosity and other necessary geophysical properties of the sample rock used for their experiments.

### 2.4.2. Pore Shape Geometry in a CT-Scan Image Sample

On the contrary, the shapes of the pores are considered to be spherical or near-spherical, and in general, their sizes are relatively smaller than those of the fractures (though not necessarily, which can be argued in various scenarios). To be a pore, in the present work, $\mathcal{H}_i$ needs to roughly satisfy $\alpha \leq \frac{\text{major}(\mathcal{H}_i)}{\text{minor}(\mathcal{H}_i)} \leq \beta$. In the current context, we kept $\alpha$ value around 0.8. The ranges are flexible, and the authors encourage the readers to modify them based on their problem definition of porosity and other necessary geophysical properties of the sample rock.

We presented the variation of the accuracy of the proposed algorithm based on the varying $\alpha$, $\beta$, $\gamma$ and $\delta \in \mathbb{R}^+$ in Table 1. The following subsections show the procedure to calculate the shape variations of any $\mathcal{H}_i$ with respect to the circle (circle variation or $\mathcal{V}_c(\mathcal{H}_i)$) and ellipse (ellipse variation or $\mathcal{V}_\mathcal{E}(\mathcal{H}_i)$) of the same perimeter. The current work classifies any region $\mathcal{H}_i$ in an input image as a pore or a fracture based on the shape variances discussed in Sections 2.4.4–2.4.6. However, before the decision, it is inevitable to identify $\mathcal{H}_i$'s, and for this we have devised an exhaustive and recursive search technique to loop over all the dark pixels and their immediate neighbors, as shown in Algorithm 2.

### 2.4.3. Finding the Contiguous Dark Pixels Using k-IN

Initially, we determine the $\mathcal{L}$ values of all the pixels in a given input image sample and classify each of them based on their respective $\mathcal{L}$ values (see Algorithm 1). Next, we locate all the dark pixels in the input image and store their relevant information (mainly location) in a separate bag of pixels, i.e., $\mathbb{L}$. Quite expectedly, the size of $\mathbb{L}$ should be any positive integer greater than 0. Next, we recursively identify all $k$-immediate dark neighboring pixels ($0 \leq k \leq 8$, see Figures 1a and 3) for every dark pixel in $\mathbb{L}$, which covers most of the dark regions in an image sample; this, in turn, points to the identification of the majority of the rock pores and fractures.

---

**Algorithm 2** Pixel-Wise *k*- IN and Classification (Pores or Fracture)

---

**Input:** Grayscale images (CT-scan) of rocks contain $l \times s$ pixels within the red boundary.

**Variables & Parameters:** $k \in \mathbb{Z}^+$; $\mathcal{L} \in \mathbb{R}^+$; $0 < I < 255$; $\alpha \leq 1$; $\beta \geq 1.3$; $0 < \gamma < 1$; and $0 < \delta \leq 1$ (for the current work).

**Objective:** Compute $\mathbb{L}_i$s and then classify them to pores and fractures.

**Output:** Pores and fractures are identified in the input grayscale images.

1: **for** ($i = 0$; $i < l$; $i$++) **do** {

2: **for** ($j = 0$; $j < s$; $j$+ +) **do** {

3: 　　**if** ($\mathcal{L}_{i,j} \leq I$) **then** $\mathbb{L} \leftarrow location(\mathcal{P}_{i,j})$ } ▷ starting pixel at $i,j = 0, 0$ (top left)

4: 　　}

5: **function 1** PIXEL-WISE_*k*-IN($\mathbb{L}$) //Primary Objective: identify $\mathbb{L}_i$

6: **for** ($i = 0$; $i <$ **area**($\mathbb{L}$); $i$++) **do** {

7: 　　$\mathbb{L}_i \leftarrow$ *k*-IN dark pixels of each $\mathcal{P}_i$

8: 　　**return** PIXEL-WISE_*k*-IN($\mathbb{L}_i$)

9: **function 2** classify Pore or Fracture //Secondary Objective: Classification of pores and fractures using the bags of pixels

10: **for** ($i = 0$; $i <$Size($\mathbb{L}$); $i$++) do {

11: 　**if**(**area**($\mathbb{L}_i$) $\neq 0$) **then**

12: 　　**for** ($j = 0$; $j <$ **area**($\mathbb{L}_i$); $j$++) **do** {

13: 　　　**for** ($r = 0$; $r <$ **area**($\mathbb{L}_{i,r}$); $r$++) **do** {$\mathcal{H}_i \leftarrow \mathbb{L}_i[r]$ } }

　　　　　　　　　▷　$\mathbb{L}_{ir}$ is the bag of pixels for $\mathbb{L}_i[r]$ and $\mathbb{L}_i[r]$ is the *r*th element of the *i*th bag of pixels, called $\mathbb{L}_i$.

14: 　　　compute the major axis length of $\mathcal{H}_i$ or major($\mathcal{H}_i$)

15: 　　　compute the major_angle($\mathcal{H}_i$)

16: 　　　compute the minor axis length of $\mathcal{H}_i$ or minor($\mathcal{H}_i$)

17: 　　　compute area($\mathcal{H}_i$) = Total number of pixels in $\mathcal{H}_i$

18: 　　　compute the perimeter of $\mathcal{H}_i$ or perimeter($\mathcal{H}_i$)

19: 　　　compute $\mathcal{V}_c(\mathcal{H}_i)$ and $\mathcal{V}_{\mathcal{E}}(\mathcal{H}_i)$

20. 　　　compactness($\mathcal{H}_i$) = $\frac{(\text{perimeter}(\mathcal{H}_i))^2}{4\,\pi \times area\,(\mathcal{H}_i)} \geq 1$

21. 　　　curl($\mathcal{H}_i$) = $\dfrac{4\times \text{major}(\mathcal{H}_i)}{\text{perimeter}(\mathcal{H}_i) - \sqrt{(\text{perimeter}(\mathcal{H}_i))^2 - 16 \times area\,(\mathcal{H}_i)}}$

22: 　width($\mathcal{H}_i$) = $\dfrac{4\times \text{area}(\mathcal{H}_i)}{\text{perimeter}(\mathcal{H}_i) - \sqrt{(\text{perimeter}(\mathcal{H}_i))^2 - 16 \times area\,(\mathcal{H}_i)}}$

23: 　　**if** (($l_p \leq$ area($\mathcal{H}_i$) $\leq \mu_p$) && ($\alpha \leq \frac{\text{major}(\mathcal{H}_i)}{\text{minor}(\mathcal{H}_i)} \leq \beta$ | | $0 \leq \mathcal{V}_c(\mathcal{H}_i) \leq \gamma$ | | $0 \leq \mathcal{V}_{\mathcal{E}}(\mathcal{H}_i) \leq \delta$)) **then** $\mathcal{H}_i$ is a pore

24: 　**else if** (($l_f \leq$ area($\mathcal{H}_i$) $\leq \mu_f$) && ($\frac{\text{major}(\mathcal{H}_i)}{\text{minor}(\mathcal{H}_i)} > \beta$ | | $\mathcal{V}_c(\mathcal{H}_i) > \gamma$ | | $\mathcal{V}_{\mathcal{E}}(\mathcal{H}_i) > \delta$)) **then** $\mathcal{H}_i$ is identified as a fracture

　　　}

25: **End**

---

For better and clearer understanding, we present an over-simplified toy example of one $4 \times 4$ image (see Figure 1). Of 16 pixels, eight among them are classified dark (classifications are done using Algorithm 1), whose locations are 1, 4, 7, 8, 10, 12, 15 and 16. We have to classify these pixels as a pore (if any), given that the collection of two or more contiguous dark pixels is considered as a pore. Initially, we kept all eight dark pixels in the bag of pixels $\mathbb{L} = \{1,4,7,8,10,12,15,16\}$ and the subsequent bags of pixels for each element of $\mathbb{L}$ are $\mathbb{L}_1 = \{\varphi\}$ (no adjacent dark pixel); $\mathbb{L}_4 = \{7,8\}$; $\mathbb{L}_7 = \{4,8,10,12\}$; $\mathbb{L}_8 = \{4,7,12\}$; $\mathbb{L}_8 = \{7,15\}$; $\mathbb{L}_{12} = \{7,8,15,16\}$; $\mathbb{L}_{15} = \{10,12,16\}$ and $\mathbb{L}_{16} = \{12,15\}$. This has been done using the recursive function PIXEL-WISE_*k*-IN($\mathbb{L}$) (see Algorithm 2).

Next, from the other bags of pixels, $\mathbb{L}_1$ is a null set, and hence pixel $\mathbb{L}_1$ cannot either be a pore or a fracture. Then for the next bag of pixel $\mathbb{L}_4$ we found 7 and 8 are the dark neighbors of pixel 4 and again from $\mathbb{L}_7$ and $\mathbb{L}_8$, it can be confirmed that $\mathcal{H}_4 = \{4,7,8,10,12\}$. After analyzing $\mathbb{L}_{10}$ and $\mathbb{L}_{12}$ (since occurring in $\mathbb{L}_7$ and $\mathbb{L}_8$), $\mathcal{H}_0$ can be extended to $\mathcal{H}_0 = \{4,7,8,10,12,15,16\}$. Later analyzing $\mathbb{L}_{15}$ and $\mathbb{L}_{16}$ since occurrence in $\mathbb{L}_{10}$ and $\mathbb{L}_{12}$) we finalized $\mathcal{H}_0$ as $\mathcal{H}_0 = \{4,7,8,10,12,15,16\}$. Since $\mathcal{H}_0$ contains more than

two pixels, hence it is considered a pore (see Figure 1a). After all the analyses, it can be confirmed that this toy example has only one identifiable pore. As mentioned earlier, it is an over-simplified case presented for a better understanding of the proposition, for which the detailed implementation of the proposed technique in the presence of a complex data set has been given in the later part to show its wide applicability range. A pictorial representation of the step-wise execution of the proposed algorithm is shown in Figure 3.

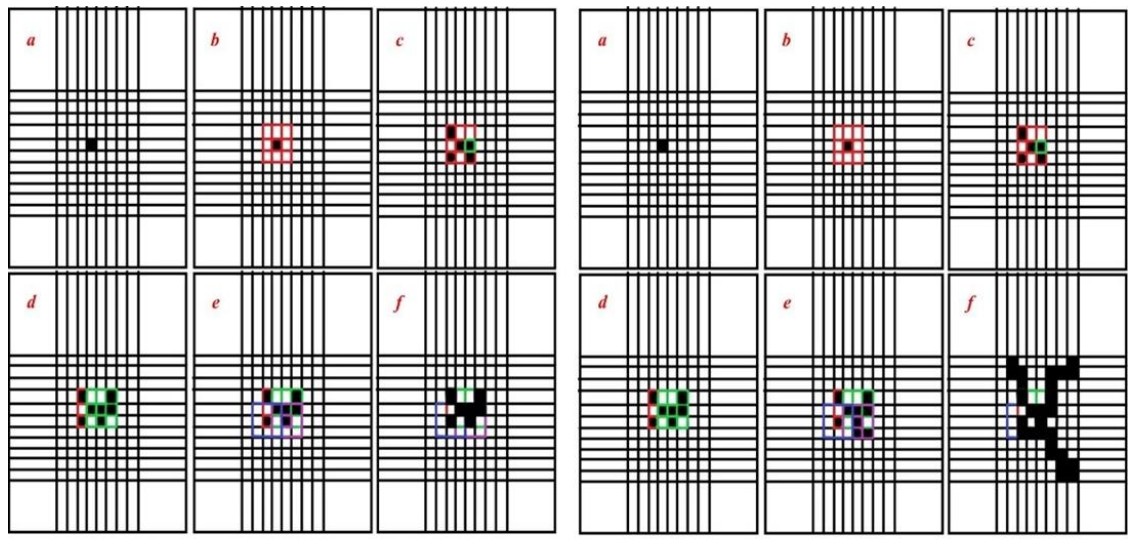

**Figure 3.** Conceptual diagrams of step-by-step from a to f each for identification of rock pores (**I**) and fractures (**II**) using the proposed *k*-IN approach. I*a.* (for simplicity) identifies one dark pixel; I*b.* identifies its immediate neighbors (marked red); I*c.* finds the immediate dark pixels in the red marked cells (in I*b.*) and points to one pixel whose nearest dark neighbors are to be found; finally, I*f.* identifies a pore by the recursive implementation of the step *b*. Quite similarly, we can detect a rock fracture in the current work. A rock fracture can also be seen as a series of connected pores in part (**II**).

### 2.4.4. Shape Variance of $\mathcal{H}_i$ with Respect to the Circle

In the current work, we have compared the shape of $\mathcal{H}_i$ with that of a circle, where the circle has the same perimeter. Perimeter ($\mathcal{H}_i$) is the number of pixels lying on its boundary. We assume there are $n$ pixels $\{x_{i,1}, x_{i,2}, \cdots, x_{i,n}\}$ lying on the boundary of $\mathcal{H}_i$ and the perimeter is computed as $\sum_{j=1}^{n-1} |x_{i,j} - x_{i,j+1}|$. We define the shape similarity of $\mathcal{H}_i$ with that of a circle as $\mathcal{V}_c(\mathcal{H}_i) = \frac{\sigma_r(\mathcal{H}_i)}{\mu_r((\mathcal{H}_i)}$, where $\sigma_r(\mathcal{H}_i)$ is the mean of radial distances from the centroid ($Gx_{\mathcal{H}_i}$, $Gy_{\mathcal{H}_i}$) of $\mathcal{H}_i$ to its boundary points and $\mu_r(\mathcal{H}_i)$ is the standard deviation of radial distances from the centroid of $\mathcal{H}_i$ to the boundary points [29,30]. Now, $\mu_r(\mathcal{H}_i)$ and $\sigma_r(\mathcal{H}_i)$ can be calculated as $\mu_r(\mathcal{H}_i) = \frac{1}{n}\sum_{j=1}^{n-1}\mathbb{D}_{i,j}$, and $\sigma_r(\mathcal{H}_i) = \sqrt{\frac{1}{n}\sum_{1j=1}^{n-1}\left(\mathbb{D}_{ij} - \mu_r(\mathcal{H}_i)\right)^2}$, respectively, where $\mathbb{D}_{ij} = \sqrt{\left(x_{i,j} - Gx_{\mathcal{H}_i}\right)^2 + \left(y_{i,j} - Gy_{\mathcal{H}_i}\right)^2}$. Now, $\mathcal{V}_c(\mathcal{H}_i) = 0$ shows the shape of ($\mathcal{H}_i$) is a perfect circle, however, with bigger $\mathcal{V}_c(\mathcal{H}_i)$ values its shape diverges from that of a circle. Hence, for a pore, we restrict $\mathcal{V}_c(\mathcal{H}_i)$ between $[0, \gamma]$, i.e., $0 \leq \mathcal{V}_c(\mathcal{H}_i) \leq \gamma \in \mathbb{R}^+$, whereas $\mathcal{H}_i$ is classified as a fracture if $\mathcal{V}_c(\mathcal{H}_i) > \gamma$ in the present work.

In the current work, we primarily varied $\gamma$-value between (0, 1), and subsequent accuracy variations of the current algorithm are listed in Table 1. The present data samples are the CT-scan micro images of carbonate rock that contain a large number of micro-pores and fractures of shapes similar to the circle (our presumption for the present case). We encourage the readers to make realistic presumptions considering the geological properties of the physical rock sample and image quality. Hence, for the best accuracy of the proposed algorithm, we decided $\gamma$-value range between (0, 0.45].

2.4.5. Shape Variance of $\mathcal{H}_i$ with Respect to the Ellipse

Next, we calculate the shape variance of $\mathcal{H}_i$ with respect to the ellipse, where the ellipse has the same perimeter (perimeter($\mathcal{H}_i$)). We calculate the shape variance $\mathcal{V}_\mathcal{E}(\mathcal{H}_i) = \frac{\overline{\sigma}_r(\mathcal{H}_i)}{\overline{\mu}_r(\mathcal{H}_i)}$, where $\overline{\mu}_r(\mathcal{H}_i)$ is defined as the mean of radial distances from the centroid $(Gx_{\mathcal{H}_i}, Gy_{\mathcal{H}_i})$ of $\mathcal{H}_i$ to its boundary points and $\overline{\sigma}_r(\mathcal{H}_i)$ is the mean standard deviation of the radial distances from the centroid of $\mathcal{H}_i$ to its boundary points. Here, $\overline{\mu}_r(\mathcal{H}_i) = \frac{1}{n} \sum_{j=1}^{n-1} d_{i,j}$ and $\overline{\sigma}_r(\mathcal{H}_i) = \sqrt{\frac{1}{n} \sum_{j=1}^{n-1} (d_{ij} - \overline{\mu}_r(\mathcal{H}_i))^2}$ where $d_{ij} = \sqrt{(W_{i,j}^T(\mathcal{H}_i) \times C_\varepsilon^{-1} \times W_{i,j}(\mathcal{H}_i)}$ information about the variance of $\mathcal{H}_i$ and the ellipse based on the radial distance, $W_{i,j}^T(\mathcal{H}_i) = x_{i,j} - Gx_{\mathcal{H}_i}, \ y_{i,j} - Gy_{\mathcal{H}_i}$ and $C_\mathcal{E}$ is the covariance matrix between the shape $\mathcal{H}_i$ and the ellipse. $\mathcal{V}_\mathcal{E}(\mathcal{H}_i) = 0$, classifies the shape of $\mathcal{H}_i$ as a perfect ellipse. Now, for $\mathcal{H}_i$ to be classified as a pore, we decided the range of $\mathcal{V}_\mathcal{E}(\mathcal{H}_i)$ as $[0, \delta]$, where $0 < \delta \leq 1$. Quite similar to the case described in Section 2.4.4, the authors encourage the readers to modify the range of $\mathcal{V}_\mathcal{E}(\mathcal{H}_i)$ depending on their problem definitions, sample rock porosity, and other necessary geophysical properties. For the best identification accuracy in the current sample images, we keep $\mathcal{V}_\mathcal{E}(\mathcal{H}_i)$ values low, and hence, we decided $\delta$-value as $0 < \delta \leq 0.45$.

2.4.6. Shape Variance Using Length and Width of $\mathcal{H}_i$

Apart from the above two measures, we also computed the ratio between the length and width of $\mathcal{H}_i$. Here, the length is assumed to be the major axis of $\mathcal{H}_i$. The major axis of $\mathcal{H}_i$ is the longest straight line drawn through $\mathcal{H}_i$ joining the endpoints $(x_{i,1}, y_{i,1})$ and $(x_{i,2}, y_{i,2})$ and its length can be computed as major$(\mathcal{H}_i) = \sqrt{(x_{i,1} - x_{i,2})^2 + (y_{i1} - y_{i2})^2}$. The angle between the major axis of $\mathcal{H}_i$ and the $x$-axis (also known as the orientation of $\mathcal{H}_i$ has been computed as major_angle$(\mathcal{H}_i) = tan^{-1}(\frac{y_{i,2} - y_{i,1}}{x_{i,2} - x_{i,1}})$. Similarly, the width of $\mathcal{H}_i$ has been assumed to be its minor axis, which is the longest line drawn through $\mathcal{H}_i$ joining its two endpoints $(x_{i,3}, y_{i,3})$ and $(x_{i,4}, y_{i,4})$ and perpendicular to the major axis. Its length is computed as minor$(\mathcal{H}_i) = \sqrt{(x_{i,3} - x_{i,4})^2 + (y_{i,3} - y_{i,4})^2}$. The major and minor axes endpoints are found by computing the pixel distance between every combination of border pixels in the object boundary and finding the pair with the maximum length, where the two straight lines are perpendicular to each other. Since the shapes of the pores are assumed to be spherical or near-spherical, we restrict the range of $\frac{major(\mathcal{H}_i)}{minor(\mathcal{H}_i)}$ between $[1, 1.3]$ ($\alpha = 1, \ \beta = 1.3$), whereas for fractures, the ratio has been kept within $(1.3, \infty)$ in the current context. To reiterate, the authors encourage the readers to modify the regions based on their problem statements, the porosity of the rock sample, and its other necessary geophysical properties.

2.4.7. Classification Logic

The present algorithm classifies one $\mathcal{H}_i$ as a micro-pore or a fracture based on its size and shape. In general, the size of a fracture is often found to be larger than that of a pore (although this proposition remains arguable in several situations). Hence, the adopted logic [31–34] to classify a pore initially checks the number of dark pixels in $\mathcal{H}_i$ (or area$(\mathcal{H}_i)$) and if the area$(\mathcal{H}_i)$ lies within a specific range $l_p \leq \text{area}(\mathcal{H}_i) > \mu_p$ (decided by the model developers) and at least one among the three $\alpha \leq \frac{major(\mathcal{H}_i)}{minor(\mathcal{H}_i)} \leq \beta, 0 \leq \mathcal{V}_c(\mathcal{H}_i) \leq \gamma$ and $0 \leq \mathcal{V}_\mathcal{E}(\mathcal{H}_i) \leq \delta$ is satisfied then we can classify $\mathcal{H}_i$ as a pore (see line no. 23 in Algorithm 2). Here, $l_p, \ \mu_p \in \mathbb{Z}^+$ are the lower and upper-bounds of area $(\mathcal{H}_i)$ respectively, values of which should be decided based on the historical information about the sizes of the pores and fractures and porosity of the original rock sample.

Again, a region containing a large number of dark pixels or $\mathcal{H}_i$ should be classified as a fracture if $l_f \leq \text{area}(\mathcal{H}_i) \leq \mu_f$ and at least one among the three conditions $\frac{major(\mathcal{H}_i)}{minor(\mathcal{H}_i)} > \beta$, $\mathcal{V}_c(\mathcal{H}_i) > \gamma$ and $\mathcal{V}_\mathcal{E}(\mathcal{H}_i) > \delta$ is satisfied (see line no. 24 in Algorithm 2). Quite similarly,

$l_f$, $\mu_f \in \mathbb{Z}^+$ are, respectively, the lower and upper-bounds of the area($\mathcal{H}_i$) for fracture. We may again repeated that the model parameters are flexible and depend on the problem definitions, sample rock porosity, and other geophysical properties.

We have also computed some other measures for a specific $\mathcal{H}_i$ such as compactness($\mathcal{H}_i$), curl($\mathcal{H}_i$) and width($\mathcal{H}_i$). Here, compactness($\mathcal{H}_i$) is an intrinsic property of $\mathcal{H}_i$ and defined as the ratio of the area of $\mathcal{H}_i$ to the area of a circle with the same perimeter (perimeter($\mathcal{H}_i$)). Mathematically, compactness($\mathcal{H}_i$) = $\frac{(\text{perimeter}(\mathcal{H}_i))^2}{4\,\pi \times area\;(\mathcal{H}_i)}$, where perimeter($\mathcal{H}_i$) is the total number of pixels residing on the perimeter of $\mathcal{H}_i$ and area($\mathcal{H}_i$) is the total number of pixels in $\mathcal{H}_i$; curl($\mathcal{H}_i$) of $\mathcal{H}_i$ is the degree to which an object is curled up. Lastly, the width($\mathcal{H}_i$) is the average width of any $\mathcal{H}_i$. These shape-specific measures can be used as features in deep-CNN, which will be considered in our upcoming research work. Visual representations of these measures are shown in Figure 1b,c.

2.4.8. Overall High-Level Architecture and Computational Complexity of the Proposed Pixel-Wise k-IN Algorithm

We felt that the overall architecture/logical flow of our proposed approach would help us explain it comprehensively. Below, we present an overall flowchart (Figure 4) combining the above steps to provide a snapshot view of our proposed algorithm: (a) images extracted from the carbonate rocks using CT Scan, (b) binarizing the pixel to dark or bright classes (c) procedure to find *k*-INs of any pixels as per Algorithm 1 (d) calculation of shape geometry KPI's such as area($\mathcal{H}_i$), major($\mathcal{H}_i$), minor($\mathcal{H}_i$), perimeter($\mathcal{H}_i$) and curl($\mathcal{H}_i$) of pore and fracture also with shape variances, (e) Classification as per the logic into pore or fracture.

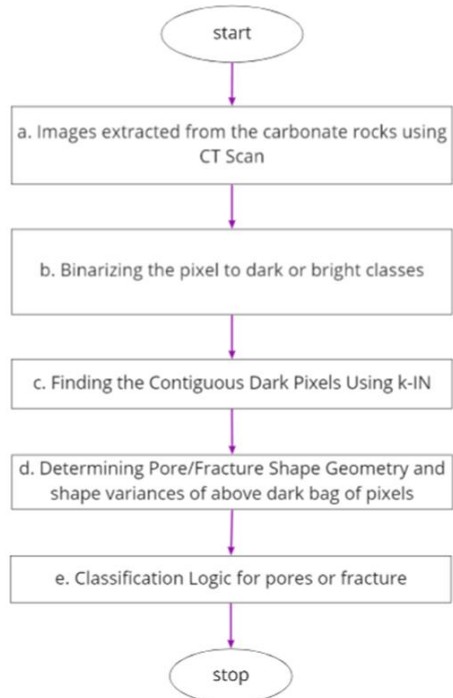

**Figure 4.** Overall architectural flow: (**a**) Images extracted from the carbonate rocks using CT Scan, (**b**) binarizing the pixel to dark or bright classes (**c**) find contiguous pixels using k-IN (**d**) procedure to find *k*-INs of any pixels as per Algorithm 1 (**d**) finding shape geometry kpi's such as area($\mathcal{H}_i$), major($\mathcal{H}_i$), minor($\mathcal{H}_i$), perimeter($\mathcal{H}_i$) and curl($\mathcal{H}_i$) of a pore and fracture and (**e**) classification into pore or fracture.

Further to the overview above, we would like to deliberate on the computational complexity of Algorithm 2. Assuming that for each operation the computer takes the same time, and *n* is the size of the input. Where image length = $\frac{n}{2}$ and breadth = $\frac{n}{2}$ assuming a

square image for the convenience of calculating computational complexity, and $n_1$ is the size of $\mathbb{L}$ (bag of pixels) $\forall\ n_1 < n;\ n_1 \subset n$. As we can see, we have mainly a few blocks of code as follows:

(a) Computing computational complexity of our proposed approach:

1.   Ingest pixels for process # runs $n$ times
2.   2 nested loop #runs $\frac{n^2}{4}$ ($\frac{n}{2} \times \frac{n}{2}$)—(line 5)
    a.   Outer for loop to iterate on the length of the image (starting LoC 1), runs $\frac{n}{2}$ times
    b.   Inner for loop to iterate on the height of the image (starting LoC 2), runs $\frac{n}{2}$ times
3.   Similarly, additional loop starting, runs $\frac{n^2}{4}$ times—(line 6)
4.   Similarly, next 3 nested loops, runs $n^3$ times—(line 10)
5.   Next steps are the calculation and if statements, runs $n_1$ times each
    a.   compute the major axis length of $\mathcal{H}_i$ or major($\mathcal{H}_i$)—(line 14)
    b.   compute the major_angle($\mathcal{H}_i$)—(line 15)
    c.   compute the minor($\mathcal{H}_i$)—(line 16)
    d.   compute area($\mathcal{H}_i$)—(line 17)
    e.   compute the perimeter($\mathcal{H}_i$)—(line 18)
    f.   compute $\mathcal{V}_c(\mathcal{H}_i)$ and $\mathcal{V}_{\mathcal{E}}(\mathcal{H}_i)$; $2n$ time—(line 19)
    g.   compactness($\mathcal{H}_i$)—(line 20)
    h.   curl($\mathcal{H}_i$)—(line 21)
    i.   width($\mathcal{H}_i$)—(line 22)
    j.   check for Pore or Fracture—(line 23)

Thus, the combined execution time is $n + \frac{n^2}{4} + n + n^3 + 11n_1$. Now we can ignore the lower order terms since the lower order terms are relatively insignificant for large inputs compared to the highest order term. Therefore only the highest order term is taken (without constant). So, the calculation complexity Big(0) will be O($n^3$).

(b). Computing the computational complexity of YOLO and RCNN approaches:

1.   In general, an MLP with $n$ inputs and $m$ hidden layers, where the $i$-th hidden layer contains $m_i$ hidden neurons and $k$ output neurons, will perform the following multiplications (excluding activation functions):

$$nm_1 + m_1m_2 + m_2m_3 + m_3m_4 \ldots + m_{M-1}m_M + m_Mk \qquad (6)$$

which in a big-O notation can be written as

$$\Omega(nm_1 + n_Mk + \sum_{i=1}^{M-1} m_im_{i+1}) \qquad (7)$$

where $\Omega$ is the lower bound and big-O is the upper bound.

2.   Add the convolution layer: Inputs: Image ($\frac{n}{2} \times \frac{n}{2}$) and convolution mask ($s \times s$). The convolution computation complexity is O($\frac{n^2}{4} \times s^2$). For simplicity, we assume $n = \frac{n}{10}$. The equation translates to O($\frac{n^4}{400}$). Dropping the constant and considering only highest-order terms again will result in O($n^4$).

3.   By putting them together, we have

$$\Omega(nm_1 + n_Mk + \sum_{i=1}^{M-1} m_im_{i+1}) + O\left(n^4\right). \qquad (8)$$

By generalising the above steps for the RCNN and YOLO, the computation complexity for both will be $\Omega(n^4)$.

The comparison of both shows that the RCNN and YOLO approaches tend to be more complex than the proposed approach.

## 3. Results and Discussion

This section briefly articulates the results of the proposed method in the presence of the pre-processed dataset shown in Section 2.3. A few output samples are shown in the image below (see Figure 5). We encircled each $\mathcal{H}_i$ considering the distance between the two farthest dark pixels of $\mathcal{H}_i$ (say, $\mathcal{P}_{i,1} = (x_{i,1}, y_{i,1})$ and $\mathcal{P}_{i,n} = (x_{i,n}, y_{i,n})$, and $\mathbb{D}(\mathcal{P}_{i,1}, \mathcal{P}_{i,n})$ is the distance between them) or major($\mathcal{H}_i$) as the diameter of the circle (see Figure 4b). We chose a yellow circle if $\mathcal{H}_i$ is classified as a pore and a white circle for a fracture (see Figure 5). Here, $\mathbb{D}(\mathcal{P}_{i,1}, \mathcal{P}_{i,n}) = \sqrt{(x_{i,1} - x_{i,2})^2 + (y_{i,1} - y_{i,2})^2} \in \mathbb{R}^+$ is the Euclidean distance between $\mathcal{P}_{i,1}$ and $\mathcal{P}_{i,n}$ (see Figure 5b).

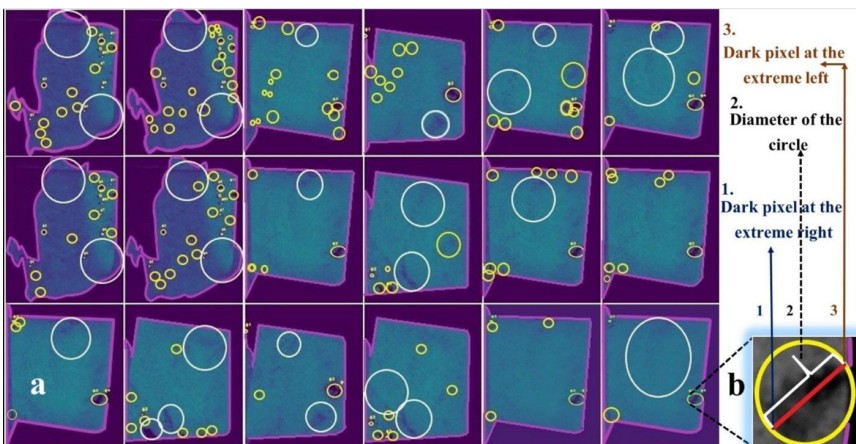

**Figure 5.** (**a**). Regions encircled in yellow are identified as pores, whereas the regions encircled with white circles are recognized as fractures. (**b**). Enlargement of one pore and its detection policy. These samples have undergone Gaussian smoothing. and hence the difference is to the naked eye.

To test the effectiveness and applicability of the proposed identification strategy (see Section 2.4), we used the pre-processed data elaborated on in Section 2.3 and confined its execution within the effective region of each image. The subsequently detected pores (encircled yellow) and fractures (encircled white) are shown in Figure 5. Here, the pores and fractures are classified based on the shape and size of $\mathcal{H}_i$ (see Algorithm 2).

### 3.1. Performance Comparison

We compared the performance of the proposed approach with that of supervised deep CNNs [18,31,35–37]. This is because we found use of the latter in recent geology literature focusing on rock porosity classification. In addition, we also compared them with industry-standard object detection models such as YOLO5 [34,38] and Faster RCNN [37]. All experimental verifications were performed using an HP-Blade server with 64-bit Ubuntu 16:04, kernel 4.4, Intel Xeon(R) CPU E5-2690 2.6 GHz (64 cores), RAM 128GB, 5.4TB SSD [32,33,37,39]. Detailed comparative criticisms are presented in the following subsections.

#### 3.1.1. Comparison with Abedini et al.

We compared our proposed study with [18], which employs deep learning involving back-propagation (network containing three layers and 30 neurons) and stacked autoencoder networks. The network structure was not clearly articulated in their study [18], i.e. without any pixel, resolution and dimension-specific information about the used image samples. Moreover, the visual evidence of detecting pores and fractures using this strategy [18] was not presented to corroborate their claim of accuracy, which certainly reduces its acceptability to the larger image-processing community. The induced strategy of [18] was to extract the shapes of pores and fractures from the image samples using 'Image Pro Plus' software and then use these extracted shapes as a training set to identify the same in the testing of high-resolution (with good chrominance) image samples containing a single

large pore or a single large fracture. The used images have good chrominance values, as the pores and fractures are marked in blue in their study. Additionally, the pores and fractures in training and testing data used in [18] are visually large and clearly recognisable to the human eye, which makes the detection mechanism relatively simple. Hence, we suspect that the identification mechanism of [18] will fail in the presence of the few low-quality grayscale images containing micro-pores and fractures in our dataset. Several reasons for this failure are possible: (i) an inadequate number of training samples for learning the necessary features to understand the shapes of pores and fractures using deep learning and (ii) extracting shape-related information from micro-pores and fractures is a tough challenge for the strategy [18].

In contrast, our proposed pixel-wise *k*-IN does not require a large image set for accurate identification. The strategy presented in [18] is mostly not applicable in the current dataset (see Section 2.3) because of its adopted learning style that requires a single pore or a fracture for training and a single pore or a fracture for testing.

### 3.1.2. Comparison with Supervised Deep CNN

Next, we compare the performance of the proposed approach (see Figure 5) with three methods of supervised deep CNN (shown in Figures 6–9), namely:

1.  Custom Object Detection model
2.  YOLOv5 and the Object Detection model

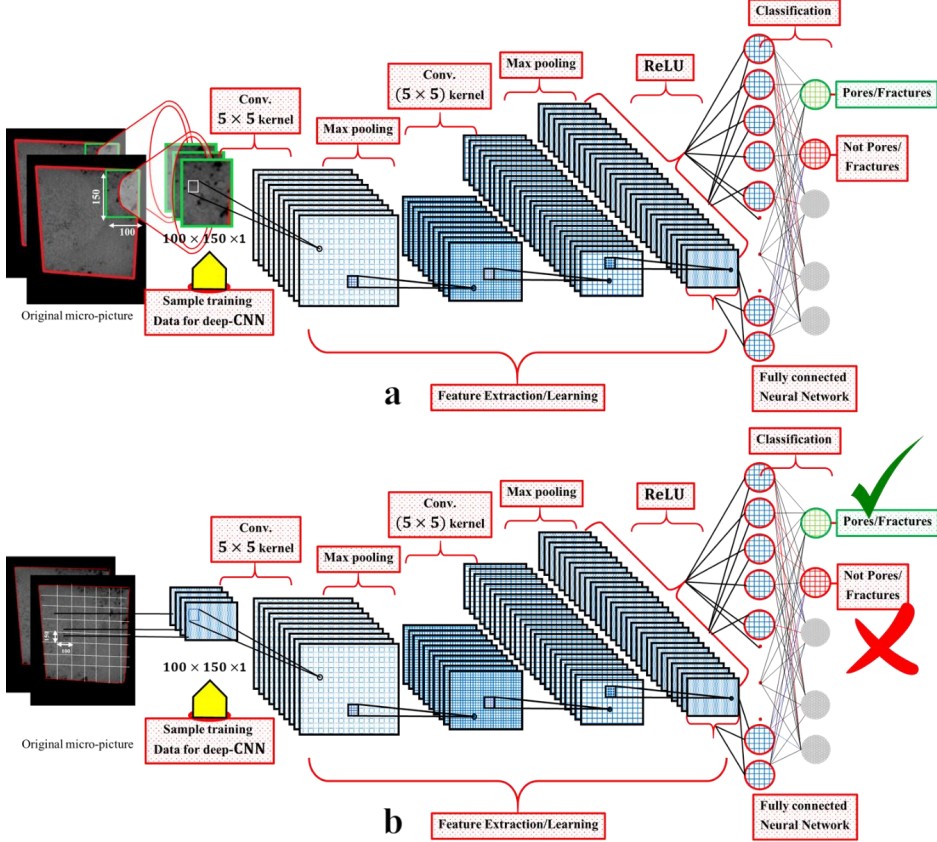

**Figure 6.** Base architecture of deep CNN adopted for various object detection experiments in the current work. (**a**) Training samples are enclosed in the green boxes of dimensions $100 \times 150 \times 1$; (**b**) For testing, we read the consecutive regions, each containing $100 \times 150 \times 1$ pixels for each testing image.

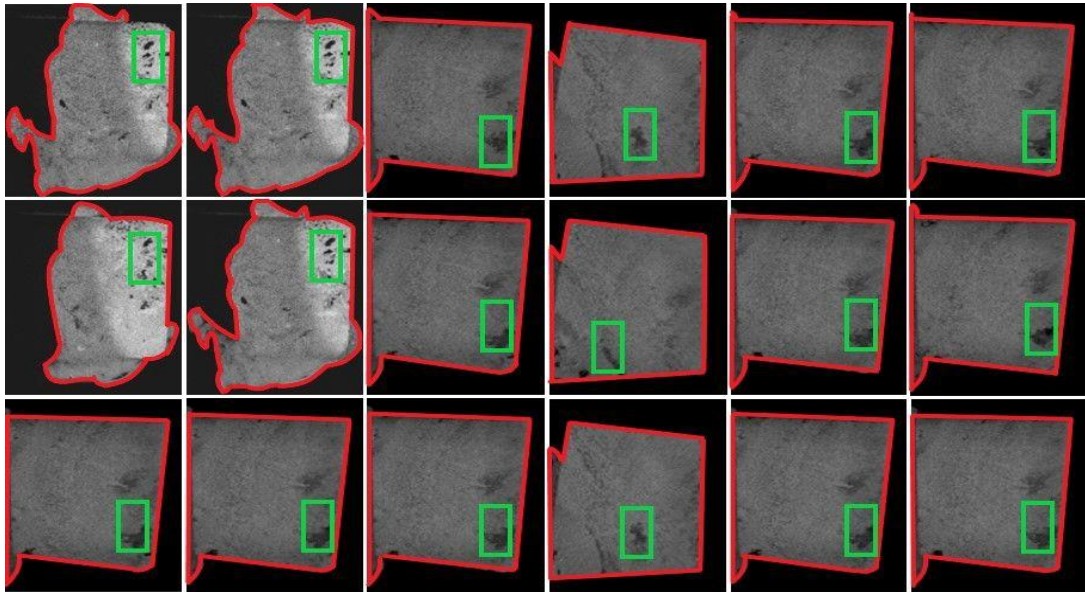

**Figure 7.** Regions enclosed by green boxes show the deep CNN model output. This figure clearly shows the lesser identification accuracy of deep CNN compared with the proposed pixel-wise *k*-IN approach.

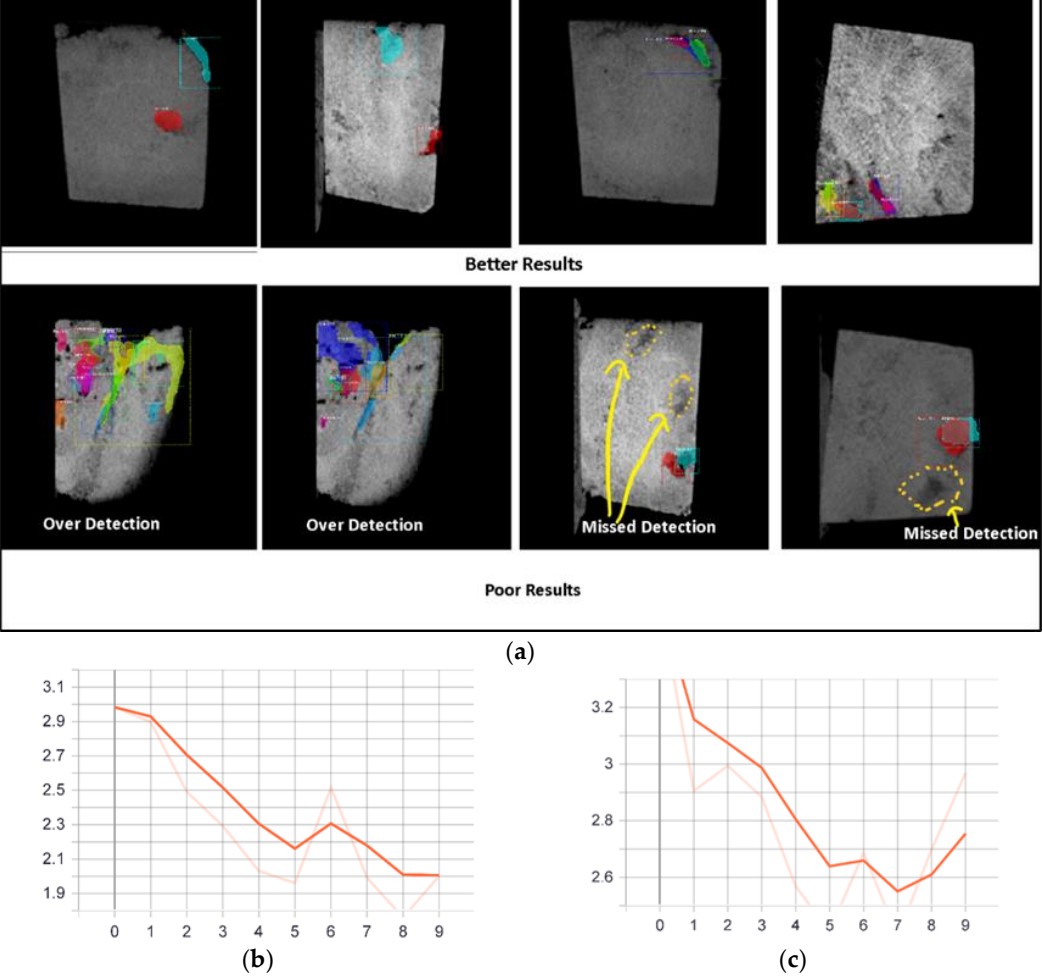

**Figure 8.** (**a**). Sample results with detected pores using deep learning mask RCNN. (**b**). Mask RCNN Train Loss by Epoch. (**c**). Mask RCNN Validation Loss by Epochs.

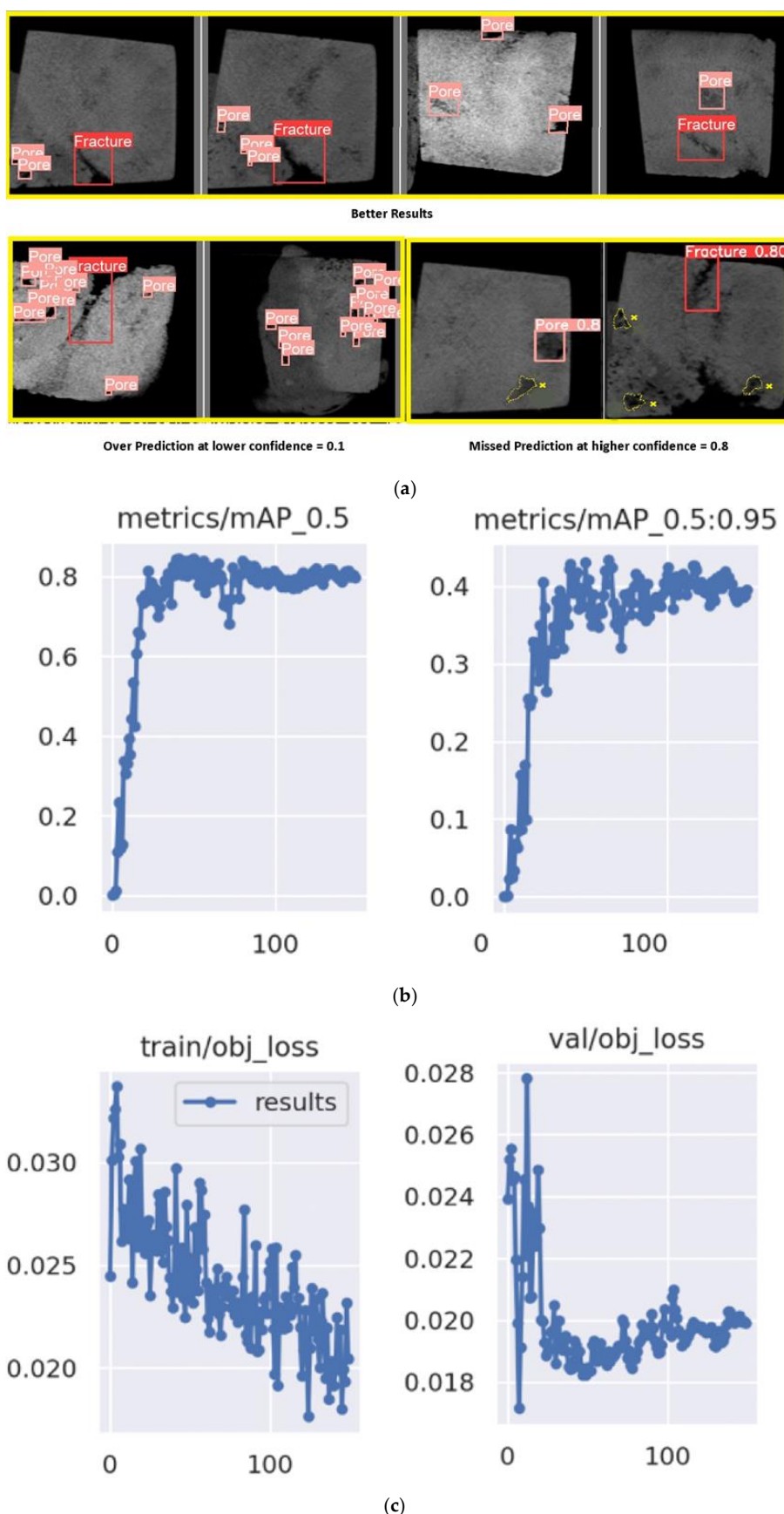

**Figure 9.** (**a**) Sample results with detected pores using Yolo v5 Deep-CNN. (**b**) mAP trend performance by confidence using Yolo v5 Deep-CNN. (**c**) Loss trend performance by confidence using Yolo v5 Deep-CNN.

Comparison with Custom Object CNN model (Figure 5):

We prepared the data in a slightly different manner. Here, we manually segmented each micro-image sample based on the many contiguous dark pixels within them. The segment size was kept fixed to $100 \times 150$ pixels, and each segment is enclosed in a green rectangle (see Figure 6a). Next, we used these segments as the training set for the Custom Object CNN. Here, we relied on the human eye to prepare the training set, whereas for testing, we use the pre-processed full micro-images shown in Section 2.3. A sample visualisation of the construction of the segments is shown in Figure 6a. Our goal is to locate the region from each micro-image from the testing set containing the most dark pixels and subsequently classify it as a pore or a fracture. Our code reads every consecutive region containing $100 \times 150$ pixels from each testing image sample and subsequently classifies it as a pore or a fracture. The resulting images from this strategy locate the regions by enclosing them with green rectangles of $100 \times 150$ pixels (see Figure 7).

To our knowledge, deep CNN models [31] have never been tested with grayscale CT scan images to identify rock pores and fractures, particularly at the micro-level. We found that the identification accuracy of the Custom Object CNN model is poorer than that of the proposed method. For Custom Object CNN models, we considered two $5 \times 5$ convolution (non-strided), max-pooling layers and fully connected layers with ReLU activation (see Figure 6). Max pool was used as a noise suppressant. It discards the noisy activation altogether and performs denoising along with dimensionality reduction and thus can be considered better than average pooling.

Comparison with the YOLO and RCNN models:

We further trained and tested industry-standard CNN networks, such as YOLO5 [34] and Faster RCNN [37,39] (see Figures 8 and 9), for object detection. These object detection models combine bounding box prediction and object classification into a single end-to-end differentiable network.

Faster RCNN, on the other hand, is the fastest member of the RCNN family. RCNN extracts a bunch of regions from a given image using a selective search and then checks if any of these boxes contains an object. We first extract these regions, and for each region, CNN is used to extract specific features. Finally, these features are then used to detect objects. Unfortunately, RCNN becomes rather slow because of these multiple steps in the process. RCNN uses selective search to generate regions. Faster RCNN replaces the selective search method with a region proposal network, from which the algorithm gets its speed much faster.

YOLO was written and is maintained in a framework called Darknet. YOLOv5 is the first YOLO model to be written in the PyTorch framework, and it is much more lightweight and easier to use. YOLO stands for 'You Only Look Once'. YOLO5 is a novel CNN that detects objects in real time (fastest) with great accuracy. This approach uses a single neural network to process the entire picture, then separates it into parts and predicts bounding boxes and probabilities for each component. These bounding boxes are weighted by the expected probability. Comparing Faster RCNN with YOLO5, we found that YOLO5 is better in accuracy for real time prediction whereas RCNN delivers better accuracy for static predictions such as video post processing as seen in Figures 8 and 9 and Table 1.

The accuracy of the abovementioned deep CNN models was quite unsatisfactory compared to that of the proposed method. This deficiency was often due to the black-box nature of deep CNN architectures, which sometimes makes them difficult to trace, verify and debug. Additionally, while handling the image samples, we often noticed very thin (mostly vague) boundaries of the pores and fractures ($\mathcal{H}_i$). This characteristic is mainly due to negligible intensity differences between the bordering pixels and the pixels residing just outside $\mathcal{H}_i$. Moreover, in many cases, we often observed negligible intensity differences among most pixels residing in a specific $\mathcal{H}_i$ and the other pixels residing in several other parts of the same image sample, which makes it very difficult for deep CNN to extract pore- and fracture-specific shape information, possibly increasing the chance of misclassification [34,37]. In contrast, models such as YOLO5 [34] and Faster RCNN [37,39]



seem to underperform because of the low quality of images and the irregular shape of pores and fractures, as detailed in the forthcoming sections.

However, we also suspect the implementation strategy of CUSTOM Object CNN which was included in the current work for comparison with the performance of the proposed approach might be the other reason behind its poor performance. This can be rectified in future work, which will be a novel contribution in the field of identifying pores and fractures of the rock from the CT-scan images.

### 3.1.3. Accuracy/Result: Process and Comparison

To compare the accuracy of the proposed method with Custom Object CNN, we manually classified each $\mathcal{H}_i$ in every image sample as a pore or a fracture after analysing its visual properties (area and shape). This classification was further supported by its geophysical properties from its mother carbonate sample rock. This dataset will be our reference for further comparison. Next, we automatically classify every individual $\mathcal{H}_i$ using the proposed pixel-wise *k*-IN and the deep CNN models and compare their results with the above reference dataset. The results, listed in Table 1, demonstrate that the accuracy of the proposed method pixel-wise *k*-IN (Figure 5) evidences better performance than the three deep CNN models (Figures 7–9). From Table 1, the accuracy of the proposed method is 85.9%. The error rate shown is due to the few observed variations in the geometric shapes of pores and fractures from the identification logic used in Algorithm 2 (see line nos. 23–24). For example, we manually identified a few micro-pores ($\mathcal{H}_i$s) after carefully analysing their visual and geophysical properties, classified incorrectly by Algorithm 2. Similarly, we manually identified a few fractures that do not satisfy the criteria shown in line no. 24 of Algorithm 2. Additionally, we showed the variation in the accuracy of the proposed algorithm by varying the model parameters $\alpha$, $\beta$, $\gamma$ and $\delta$ in Table 1. This action also shows that the correct combination of parameters for the current dataset is $\alpha = 1.03$, $\beta = 0.95$, $\gamma = 0.43$ and $\delta = 0.43$.

In contrast, the error rate of the industry-leading deep CNN models (YOLO and RCNN) was due to the poor learning of different shape-related features and the fewness of training samples. The YOLO model provides accuracy that is based on the intersection of actual bounding boxes and predicted by the feedforward network. Since the predicted bounding box will not match the ground truth (see Figures 7 and 9a), we obtain a ratio by dividing the area of intersection between the two boxes by the area of their union, which is called the intersection of unit. The higher this ratio is, the better the prediction. By equating this prediction with the ground truth, we obtain the accuracy of the model.

The possible reasons for the better performance of the proposed approach in the presence of the current image samples are elaborated on in the subsequent subsections.

### 3.2. Reasons behind the Failures of the Contemporary Approaches

This section briefly discusses the possible reasons behind the failures of the modern contemporary approaches (deep CNN models) and the several advantages of the proposed algorithm in detecting rock pores and fractures by analysing their CT scan images. One of the most important advantages of the proposed technique is its simplicity of implementation and easy understanding. The remaining advantages are based on structural/conceptual comparisons with the existing techniques in a similar domain.

### 3.2.1. Unavailability of High-Quality Data

Image quality is often considered an important, frequently faced challenge in machine vision systems. Image quality refers to the high resolution and chrominance of the image. Commonly, machine vision systems are trained and tested on a high-quality image dataset, yet the high quality of the dataset cannot always be assumed [40]. For this reason, several contemporary machine vision algorithms underperform in the real world. Quite expectedly, we also found that the detection quality of a deep CNN model's performance strongly depends on the quality of the samples.

However, as discussed earlier, the proposed pixel-wise k-IN is robust in this situation because of its exhaustive search mechanisms to locate the dark pixels and their k-immediate dark neighbours deterministically. This approach helps pixel-wise k-IN identify the rock pores and fractures in each image sample. Other model parameters, such as $\alpha$, $\beta$, $\gamma$ and $\delta$, are responsible for classifying $\mathcal{H}_i$ as a pore or a fracture. The values of the model parameters depend on the problem statement, sample rock porosity and its other necessary geophysical properties.

### 3.2.2. Samples with Less Divergence

Low divergence samples accompanied by a small number of grayscale image samples constitute another issue we encounter in achieving accuracy using conventional methods. The dataset used in the current work has considerably less divergence, i.e., the visual quality (the location and shape of $\mathcal{H}_i$) of most images in the sample is very similar. The reason behind the visual similarity is given in Section 2.3, and a method for measuring the similarity between the image samples is provided in Section 2.2. Even in the presence of a formidably large number of grayscale image samples, we found deep-CNN models (including ours) were failing to identify the regions containing the pores and fractures with an accuracy as high as in Figure 10 [18,25]. This failure could be due to the poorly crafted training samples (having fewer unique images) leading to a mode-collapse problem, i.e., being stuck in local minima because of poor learning.

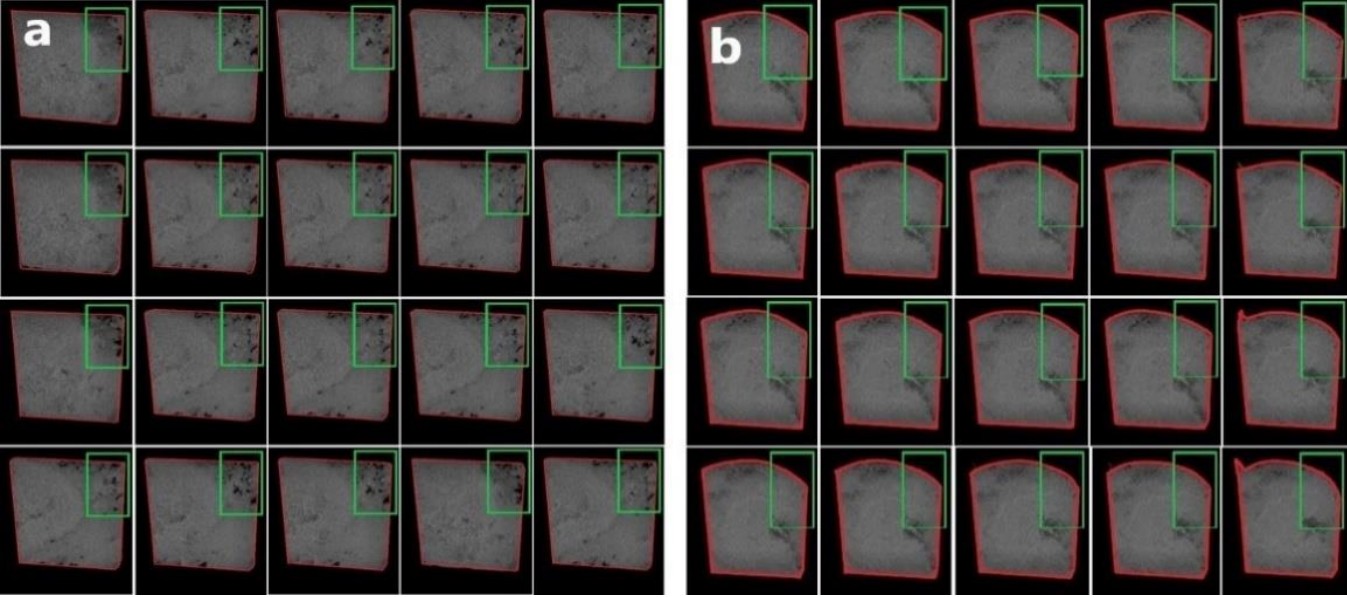

**Figure 10.** Evidence of location learning using Custom Object CNN: (**a**) Sample CT scan images with manually identified regions containing the possible pores and fractures of the rock used as training samples; (**b**) In return, the Custom Object CNN identifies the same regions (as in (**a**)), eventhough very few pores are located. This case is a classic example of mode collapse. One possible solution to the mode-collapse problem is minibatch learning; however, this approach might fail in the absence of a large and diverse sample image dataset. To combat this problem, we increased the diversity while preparing the training samples for Custom Object CNNs, but this policy fails because of the lack of variety within the current dataset. In figures, (**a**,**b**) the image boundaries are marked in red.

To combat this problem, we conducted experiments using the Custom Object CNN model in two ways: (i) Learning manually identified regions containing the maximum numbers of micro-pores and fractures as training samples. We maintained higher visual dissimilarity while preparing the training samples. A comparative study considering the training sample divergence is shown in Figure 11. (ii) Learning the geometric shapes of individual micro-pores

and fractures to identify them correctly in the test samples. The manually identified regions of several images containing the most pores and fractures are chosen as training samples in case (i), whereas the normal shapes of the pores and fractures are taken as training samples for the latter case (ii). Each experiment strategy is described below.

1. Manually identify the regions containing the most possible pores and fractures: For this task, initially, we manually encircled (in green boxes) the regions containing the most possible pores and fractures to create a training set containing 300 images (see Figure 10a). We then used this training set to train the Custom Object CNN model in Section 3.1.2 and used the remaining images as a test dataset. High visual similarity can be observed among different grayscale images of the current sample, triggering a mode-collapse problem that forces the Custom Object CNN model to stick to a suboptimal solution due to the high degree of similarity among the extracted features. Figure 10b shows the evidence of mode collapse.

    a. The training image samples in Figure 10a have visible pores at specific locations, mainly at the corners (enclosed in green boxes in Figure 10a). Initially, these samples were used to train the Custom Object CNN model, and a part of the resulting predictions is shown in Figure 10b, where we found that the model can identify only the learnt locations in the resulting images. For example, using testing samples with pores located in the middle, we found that the deep CNN failed to identify these pores (see Figure 10b). We suspect that this failure occurs because the deep CNN was only trained to learn the locations of the pores and fractures situated at the corners of the image samples (mainly, the locations of the green boxes) and can only detect the pore and fracture locations at the corners of the testing images; hence, it is a case of mode collapse. This failure could also be due to the unusual dissimilarity between the training dataset locations, shapes, sizes and orientations of the pores and fractures (Figure 10a) and the testing dataset (Figure 10b). Moreover, the divergence between the training and the testing sets was significant, whereas the divergence within the training and testing sets was quite low. These findings might point to the existence of visually similar images in the training ($\mathcal{E}$) set as well as in the testing ($\mathcal{J}$) set, i.e., the images of $\mathcal{E}$ and $\mathcal{J}$ are highly statistically and visually dissimilar. For example, the average similarity/dissimilarity between the images of the training set ($\mathcal{E}$) shown in Figure 10a is quite low, with $\mathcal{D}_{\mathrm{KL}}(\mathcal{E}_i||\mathcal{E}_r) \approx 0.799$, where $\mathcal{E}_i, \mathcal{E}_r \in \mathcal{E} \subseteq \mathcal{M}$; $i \neq r$ (see Section 2.2). Hence, the Custom Object CNN model learnt the locations of the pores and fractures of the training set and finally failed to identify the pores and fractures in the testing samples (see Figure 10b). Additionally, the average similarity/dissimilarity between the images in $\mathcal{E}$ and $\mathcal{J}$ is calculated to be quite high $\mathcal{D}_{\mathrm{KL}}(\mathcal{E}_i||\mathcal{J}_r) \approx 9.89$, where $\mathcal{E}_i \in \mathcal{E}$ and $\mathcal{J}_r \in \mathcal{J} \subseteq \mathcal{M}$. This result could be another reason behind the poor performance in the current case and might be extremely specific to the current problem in the present dataset.

    b. To combat these problems, we prepared a training set incorporating more statistically dissimilar images (partially shown in Figure 11a) and found better detection accuracy than the previous one, although still unsatisfactory (see Figure 11b). This performance improvement is mainly due to the diversity of the training samples, which helps the Custom Object CNN model to learn about more diverse locations of the pores and fractures (the locations of the green boxes), situated in the corners, middle and several other portions in the micro-images. In this case, the average dissimilarity within $\mathcal{J}$ was increased to $\mathcal{D}_{\mathrm{KL}}(\mathcal{J}_i||\mathcal{J}_r) \approx 4.799$ (higher than the previous case) while the average dissimilarity between the images of $\mathcal{E}$ and $\mathcal{J}$ was reduced to $\mathcal{D}_{\mathrm{KL}}(\mathcal{E}_i||\mathcal{J}_r) \approx 6.193$; $\mathcal{E}_i \in \mathcal{E}$; $\mathcal{J}_r \in \mathcal{J}$; $i, r \in \mathbb{Z}^+$. Quite expectedly, the resulting identification accuracy was better than the previous attempt (see Figure 10). In the current study, however, it is impracticable to increase the dissimilarity within the training samples

due to the limited size of the image sample and experimental constraint of sequential images. This result aligns with the well-accepted notion that deep learning needs a large dataset to achieve its famed accuracy.

c.  Another reason for the poor identification accuracy of the Custom Object CNN model could be the irregular shape-related features of any $\mathcal{H}_i$ from the training dataset. The irregular shapes of all $\mathcal{H}_i$'s make it difficult for model developers to identify the suitable supervised deep CNN features for their accurate classifications.

    ✓    However, this problem might be handled well using capable unsupervised deep CNN counterparts, which can automatically and intelligently identify the features required to recognise the irregular objects with higher accuracy. Custom Object CNN's poor identification accuracy may also be due to its present implementation architecture. However, this architecture is closely related to similar studies conducted on geophysical image-processing detection [18,25]. Additionally, we have compared the best-in-class object detection models in Table 1 and found results inferior to our proposed method. Hence, we firmly believe that the identification accuracy cannot be improved any further in the absence of large samples and high-quality training image samples.

d.  We also tested with the current input samples after converting the grayscale images to two predefined colours, reducing the information within the image from 256 grey shades to only 2. This procedure is known as binarization, and it often provides sharper and clearer contours of different objects in an input image [36]. However, the resulting accuracy was still found to be unsatisfactory because of the tiny sample size and unidentifiable shape-related features, making it difficult to train the Custom Object CNN model to accurately identify the irregular-shaped micro-pores and fractures. In contrast, the proposed pixel-wise *k*-IN shows satisfactory accuracy in the presence of binarised input.

2.  Learning the shapes of an individual pore and a fracture [18]: For this task, we extracted the images of the $\mathcal{H}_i$ from 400 micro-image samples as described in the work of [18] and created a training set of five shapes (such as intra-particle, vuggy, moldic, biomoldic and fracture [18]) with 50 examples of each category for the deep learning models for feature extraction and found considerably low detection accuracy because of the high geometric and visual similarity (insignificant dissimilarity) in the shapes of different types of pores and fractures in the used micro-image samples. The extracted micro-pores and fractures have inferior quality and remarkably similar shapes because of their tiny sizes. Moreover, the quality of the used grayscale image samples prevents the 'Image Pro Plus' software from extracting the high-resolution images of pores and fractures that could further be used as training samples for the Custom Object CNN. Consequently, the Custom Object CNN model fails to extract the above features that help perform the porosity classifications and fracture identifications uniquely. We found that the training grayscale image samples of micro-pores and fracture images (extracted using the 'Image Pro Plus') are highly distorted (irregularly shaped geometry) in most cases and subsequently, extracting their shape geometry-related features became extremely difficult in reality for CNN-based object detection systems, which are square bounding-based detection, whereas pores are generally irregularly shaped. These pores can sometimes be much smaller than the size of the convolution filters chosen, which leads to false positives. Hence, the identification accuracy of this strategy, only approximately 10%, is very inferior to that of case (i). In contrast, the proposed approach shows better accuracy than the above deep learning approaches because of the benefit of its pixel-level in-depth analysis and the adoption of an exhaustive search approach to identify new neighbouring dark pixels of a detected dark pixel.

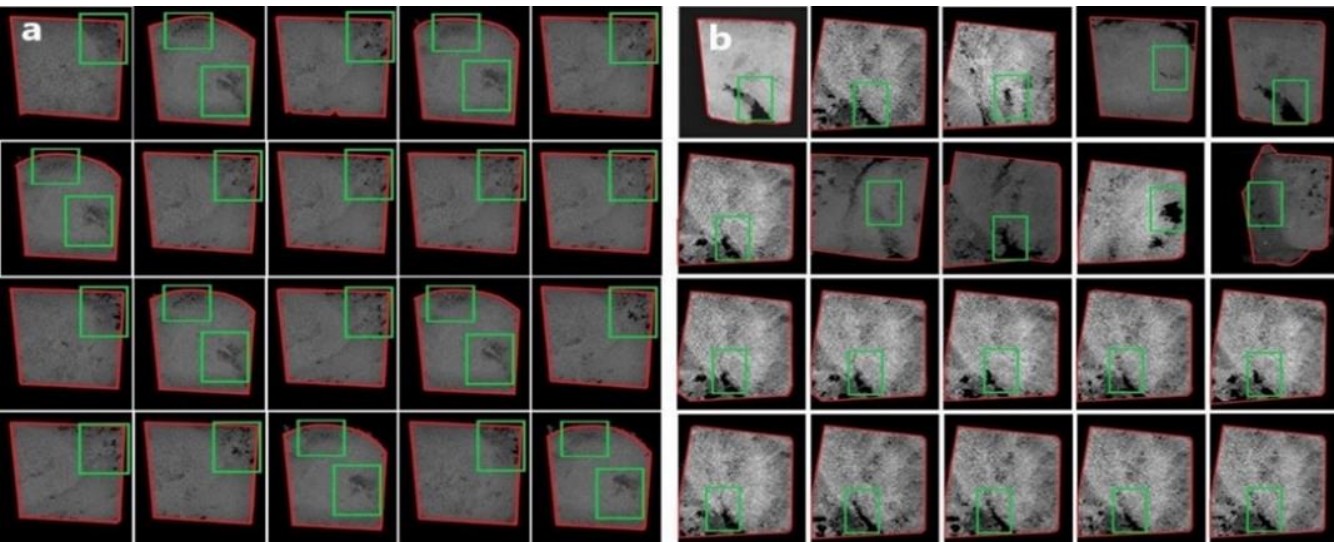

**Figure 11.** (**a**) New sample training set with the average similarity between the images calculated as $\mathcal{D}_{KL}(\mathcal{E}_i||\mathcal{E}_r) \approx 4.799$. Here, the regions containing pores and fractures are marked with green boxes. (**b**) The sample resulting images show the regions identified as pores and fractures by the deep CNN model. The identification accuracy was higher than that presented in Figure 7.

### 3.2.3. Sample with an Extremely Small Training Set

The unavailability of large and diverse set of training samples is widely considered a major practical problem behind the unsuccessful implementation of the Custom Object CNN [33,41,42]. In addition, we could not find any public dataset similar to our study to enhance the sample set size. Large datasets are often needed to ensure that the deep CNN delivers the desired results. The success of deep learning methods [18,25] is often found to entirely depend on the availability of a large and diverse set of training samples. Particularly in the present scenario, the size of the dataset is unsatisfactory; hence, the accuracy of the current Custom Object CNN, including the YOLO and RCNN models, was inferior to the proposed approach. We firmly believe that the accuracy can barely be improved in the absence of a large set of training samples. Similarly, the other deep learning implementation [18] will not be successful without a large set of training samples.

This problem has been successfully countered using the proposed approach of analysing individual images based on pixel intensity. The proposed approach does not require a learning phase and hence can successfully and accurately be applied in the presence of a tiny number of grayscale image samples.

### 3.2.4. Detecting Objects of Irregular Shape

This problem could be a major difficulty for the deep CNN approaches and [18] the tiny size of inferior quality (grayscale, resolution) images containing micro-pores and micro-fractures (visually small, in the current case). As discussed above, thin boundaries of pores and fractures are frequently observed along with a negligible level of intensity differences among the pixels lying inside or out of any tiny (small) $\mathcal{H}_i, \forall i \in \mathbb{Z}^+$ in any image sample in the present dataset (see Section 2.3).

As a result, the pores and fractures are objects of extremely unusual shapes, and accurately retrieving their shape-related relevant information (for their future identification) using the present deep CNN implementation becomes quite imperfect, thus increasing misclassification. This result could be due to the layers of deep CNN extract better and specific features related to identifying pores and fractures. Hence, to identify unusual-shaped objects such as pores and fractures (in the current situation), even at the micro-level, their features must be firmly defined before classification. This task is quite difficult in the current case because of the unusual shapes of the pores and fractures, thus the number

of misclassification events increases. When using deep CNN and [15], the number of misclassifications increases even more if the training samples are few. In contrast, the proposed approach can detect unusually shaped pores and fractures with higher accuracy. This ability is a benefit of the pixel-wise exhaustive search (by analysing the intensity) approach in every image sample and the flexible choosing of the I -value (depending on the image quality and problem requirements) adopted by the proposed procedure, which is documented in Table 1. In contrast, as mentioned before, the proposed method does not require training and hence can work well with a few datasets.

### 3.3. Advantages of the Proposed Algorithm

From the above sections, the advantages of the proposed algorithm are quite evident and listed as follows:

- The proposed technique is relatively easy to understand and implement. In addition, its identification accuracy meets our expectations for the present dataset.
- It can be successfully applied to a few grayscale image samples, whereas deep learning approaches such as Custom Object CNN, YOLO and RCNN suffer from poor accuracy in the presence of a small image dataset. These famous image classification models are trained on massive datasets. Among these datasets, the top three used for training are as follows:
  - ○ ImageNet—1.5 million images with 1000 object categories/classes,
  - ○ Microsoft Common Objects in Context (COCO)—2.5 million images, 91 object categories,
  - ○ PASCAL VOC Dataset—500K images, 20 object categories.
- The proposed algorithm can be successfully applied to a small image dataset, such as 300 images, because it does not require training for object identification.
- Further referring to our limitations of the size of the dataset and quality of the images, this technique can use samples with a low divergence and lower rank hardware resource for computing on the edge. As we know, deep CNN models depend on hyper-parameter choices such as the filter size, regularisation chosen and quantisation levels [38,42,43]. Some of the hyper-parameters we finetune in deep learning methods that can impact the accuracy gain and robustness of the models are epochs (min 1), batch size (min 1), loss function (cross entropy, L1 loss, mean square loss, negative likelihood), optimiser algorithm. learning rate, weight decay, rho, lambda, alpha, epsilon, momentum and learning rate decay. These parameter (more than 15) settings may have an infinite combination and may lead to another combinatorial optimization problem in this case [43], 320 random hyper-parameter settings (in this case) and hence it can swing an average random accuracy by as much as $41.8 \pm 24.3$ [43]. This result shows the amount of variation parametric choices can have over the results. Such high-dimension problems are then reduced in trials with extensive domain expertise, which again becomes very close to an empirical approach such as ours. In comparison, the proposed model has less than five parameters ($\alpha$, $\beta$, $\gamma$, $\delta$, *and I*) to optimise in our proposed approach. Hence, it is a much simpler and straightforward comparison.
- We have tried deep learning in our separate experiments on the Advance Drive Assistance System (ADAS) and for various COVID-19 related apps that run deep learning on-the edge computing. We found that the accuracy levels have drastically reduced the inference of the full Resnet and YOLO-based object detection model after model quantisation. There is a limit on the capacity of deep learning models, whereas our proposed model can easily make inferences on edge (IoT) hardware such as Jetson Nano and even Raspberry Pi.
- Our proposed model can be used to identify objects without fixed geometric shapes and sizes (such as pores and fractures). In contrast, deep CNN models are inefficient in this scenario because of an insufficient understanding of the necessary shape-related features required to classify any correctly.

- Furthermore, though being scientifically regressive, back-propagation (BP) with the gradient descent (GD) has the inherent issue that it frequently gets stuck in local optima [44]. This result was further supported by Gori et al. [45], who found that BP with GD can be guaranteed to find global minima only if the problem is linearly classified. As a learning algorithm, BP, unless supported by empirical techniques such as adaptive learning rate, momentum, noise randomisation and weight spawning, cannot find guaranteed global minima using current deep learning techniques, as discussed above. Furthermore, it is necessary to investigate the hyper-parameters of shallow layer perceptron to show the existence of several local minima and saddle points [46]. Again, this problem can be resolved with the help of choosing the right kind of hyper-parameter settings or some heuristics. To summarise, the issue with BP and GD is that they are slowly converging and may get stuck in local minima and saddle points indefinitely. Furthermore, it is computationally very expensive (calculate derivatives) being its inherent disadvantages, which tie back to its above-discussed limitation to deploying on-the-edge equipment. Therefore, we have proposed a lightweight method with a minimum adjustments of parameters, and we can obtain better results given the above-specified limitations.
- In addition, the computing resource requirement for the proposed approach is relatively less stringent as for manual involvement during the feature extraction process; only CPUs are sufficient to do the inferencing. In contrast, higher computing resources are needed in terms of training and inferring from deep learning experiments [47–49]. Training a deep neural network is very time-consuming. We need dedicated hardware (high-powered GPUs, high RAM, SSD, etc.) [50] to train the latest state-of-the-art image classification models in a day and on top of it in case undesirable results imagine retaining the model Imagine this issue with our attempt to create a robotic instrument using computing on the edge which can make real-time inference for pores and fractures with some empirical methods specific accuracy to the domain. We found that the accuracy levels drastically reduced the inference of the full RCNN and YOLO-based object detection model after we quantised [49] the model. Therefore, there is a limit on productionising deep learning models, whereas we found that our proposed model can easily make inferences on edge (IoT) hardware such as Jetson Nano and even Raspberry Pi.

*3.4. Restrictions of the Proposed Algorithm*

Despite having several advantages, the proposed algorithm has a few noteworthy restrictions that are listed below:

- The proposed method should be chosen only in the absence of a large and high-quality training set with considerable similarities and dissimilarities among the sample images. This restriction is given not because of its accuracy but because of its high computational time and adopted pixel-wise exhaustive image analysis mechanism. The proposed approach processes each image, and hence, its computation time might substantially increase in the presence of a large and high-resolution image dataset, although the identification error should remain the lowest [35–37]. However, in the presence of a large training set, deep CNNs might also suffer from high computational costs.
- Sometimes, in the presence of CT scan images containing micro-pores and fractures, it can be difficult to choose suitable $I$, $\alpha$, $\beta$, $\gamma$ and $\delta$ values, where we must rely on expert judgement. However, the same problem remains in deep CNN, where we often need to decide on several model hyper-parameter values.
- Multi-objective algorithms pose higher complexity and computation in comparison to their single-objective counterparts.

## 4. Conclusions

The imaging, accurate identification and modelling of pores and fractures have become flagship programmes in the leading oil and gas industries because of their immense

applications in contaminant transport and $CO_2$ storage, and often, porosity is considered a basic parameter for reservoir characterisation. However, quite frequently, the identification procedure of pores and fractures from the image samples is manual or inaccurate if automated. Above all, micro-pores and fractures in a grayscale rock image (CT scan samples) are mostly unidentifiable or indistinguishable using the contemporary techniques of automatic identification.

Prompted by this fact, in the current work, we developed a novel pixel-wise multi-objective image analysis strategy to automatically identify the pores and fractures from the input CT scan images of rocks. We have tested the applicability and efficiency of the proposed technique using CT scan images of carbonate rocks, and we firmly believe that the proposed methodology will retain its success in the presence of the CT scan images of various other types of rock samples. This belief is held because of its adopted pixel-wise intensity analysis mechanism, which enables the algorithm to understand the largest intensity variations among different pixels along with its neighbouring pixels and accordingly, classify them as bright or dark, thus helping to identify pores and fractures in the CT scan images. However, an accurate understanding of the intensity variation among the pixels in an input image sample depends on a perfect selection of $\alpha$, $\beta$, $\gamma$, $\delta$ and $I \in R+$, which requires expert judgement ability of the model developer. Next, to juxtapose the accuracy of the proposed methodology, we compared its performance with that of deep CNN and the method developed by [15] using the same image samples. The results clearly show a better identification accuracy of pores and fractures by the proposed approach using the input CT scan images.

- We identified a possible reason for the poor identification accuracy of supervised deep CNNs as the unidentified necessary shape-related features of any $\mathcal{H}_i$, which are highly required during the training of supervised deep CNNs to accurately classify any specific $\mathcal{H}_i$ as a pore or fracture. Particularly, the irregular shapes of all $\mathcal{H}$ make it difficult for model developers to identify the suitable supervised deep CNN features for their accurate classifications.
- However, this problem might be handled well using capable unsupervised deep CNN counterparts, which can automatically and intelligently identify the features required to recognise irregular objects with higher accuracy. This possibility will be investigated in our subsequent research works. The poor identification accuracy of deep CNN might be due to its present implementation, although we firmly believe that the accuracy can barely be improved in the absence of large and high-quality training samples.
- Possible Implementation of the Proposed System: The proposed system can be prepared as a module, attached to the imaging device and linked with its guiding software to simultaneously identify the pores and fractures (even at the micro-level) during the imaging of rock samples. Gathering many meaningful real data samples has always been a major problem in any real-world rock imaging scenario, which often has a major negative impact on the identification accuracy of deep learning contemporaries. Therefore, a deep CNN module, if attached to a subsequent imaging device and its guiding software as an add-on, should be pre-trained because of the lack of relevant training samples. However, the biggest challenge for the pre-trained models for understanding objects without fixed geometric shapes (such as pores and fractures) is perfectly defining and learning the necessary shape-related features that distinguish the pores from the fractures. This approach might increase the number of misclassifications for a deep CNN model. In contrast, the proposed method only requires an accurate range of its parameters such as $\alpha$, $\beta$, $\gamma$ and $I$ to uniquely identify pores and fractures. The accuracy of the proposed method was corroborated by various experiments presented in the current work.

**Author Contributions:** Conceptualization, J.W. and A.R.; methodology, A.R., P.S.N. and J.W.; software, P.S.N., A.R. and I.B.A.A.; validation, A.R., P.S.N. and J.W.; writing, P.S.N., A.R. and J.W.; visualization, P.S.N., A.R. and J.W.; supervision, A.R., J.W. and I.B.A.A.; funding acquisition, J.W. and I.B.A.A. All authors have read and agreed to the published version of the manuscript.

**Funding:** This work is partially supported by the PRF 06B grant awarded to Dr. Eswaran Padmanabhan at the Universiti Teknologi Petronas (UTP), Malaysia. We are also thankful to Dr. Eswaran Padmanabhan for providing us with the necessary data for the present research. This research is also partly supported by the Funding Name: "Predicting Missing Values in Big Upstream Oil and Gas Industrial Dataset Using Enhanced Evolved Bat Algorithm and Support Vector Regression," under the Grant Name: YUTP-FRG 1/2021 (015LC0-353).

**Institutional Review Board Statement:** Not applicable.

**Data Availability Statement:** The data presented in this study are available on request from the corresponding author. The data are not publicly available due to privacy reasons.

**Conflicts of Interest:** The authors declare no conflict of interest.

## Nomenclature

| | |
|---|---|
| $\mathcal{G}_n$ | $n$th grayscale image sample from the dataset $\mathcal{M}$ |
| $\mathcal{P}_{i,j}$ | Any pixel of location $(i, j)$ in $\mathcal{G}_n$ |
| $location(\mathcal{P}_{i,j})$ | Location of $\mathcal{P}_{i,j} \in \mathcal{G}_n$ |
| $\mathcal{L}_{i,j} \in \mathbb{Z}^+$ | Intensity of $\mathcal{P}_{ij} \in \mathcal{G}_n$ |
| $I$ | Threshold value ($\in \mathbb{Z}^+$) needed to classify $\mathcal{P}_{i,j}$ as bright or dark. It depends on the image sample quality and the developer's expert judgment |
| $\mathcal{H}_i$ | Set of contiguous dark pixels in $\mathcal{G}_n \in \mathcal{M}$, which will be classified as a pore or a fracture if it contains a predefined number of dark pixels and satisfies preconceived classification logics. Here, the size depends on the image quality and geophysical properties of the rock sample. |
| $major(\mathcal{H}_i)$ | Major axis of the region $\mathcal{H}_i \in \mathcal{G}_n$ |
| $minor(\mathcal{H}_i)$ | Minor axis of the region $\mathcal{H}_i \in \mathcal{G}_n$ |
| $area(\mathcal{H}_i)$ | Total number of pixels in $\mathcal{H}_i \in \mathcal{G}_n$ |
| $perimeter(\mathcal{H}_i)$ | Perimeter of the region $\mathcal{H}_i \in \mathcal{G}_n$ |
| $\mathcal{V}_c(\mathcal{H}_i)$ | Shape similarity of $\mathcal{H}_i \in \mathcal{G}_n$ with a circle of $perimeter(\mathcal{H}_i)$ |
| $\mathcal{V}_\mathcal{E}(\mathcal{H}_i)$ | Shape similarity of $\mathcal{H}_i \in \mathcal{G}_n$ with an ellipse of $perimeter(\mathcal{H}_i)$ |
| $\mathcal{D}_{\mathrm{KL}}(.||.)$ | KL-divergence between two distributions |
| $\mathcal{E}, \mathcal{J} \subseteq \mathcal{M}$ | Training and testing sets for deep-CNN, respectively |
| $\mathbb{L}_i$ | Set of subsequent dark pixels of any pixel in the image |
| $l_p, \mu_p \in \mathbb{Z}^+$ | The lower and upper-bounds of area $(\mathcal{H}_i)$, respectively |
| $i.i.d$ | A collection of random variables which is independent and identically distributed. |

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
