# Peer review of "A Pixel-Wise k-Immediate Neighbour-Based Image Analysis Approach for Identifying Rock Pores and Fractures from Grayscale Image Samples"

_algorithms, doi:10.3390/a16010042_

Round 1
Reviewer 1 Report
This work proposes a novel meta-heuristic image analysis approach using multi-objective optimization, named “Pixel-wise k-Immediate Neighbors,” to identify pores and fractures (both natural and induced, even in the micro-level) in the wells of a hydrocarbon reservoir, which presents better identification accuracy in the presence of the gray-scale sample rock images. proposed system can be prepared as a module, attach with the imaging device and linked with its guiding software for simultaneously identifying the pores and fractures (even in the micro-level) during the imaging of rock samples.
Overall, the problem and method have been described in much detail.
I have some concerns:
1. Cited papers in references list are quite old, may consider citing recently published related works.
2. The authors list Reasons Behind the Failures of the Contemporary Approaches (e.g., Deep CNN): Unavailability of High Quality Data, Samples with Less Divergence, Sample with Extremely Small Training Set and so on. However, these issues can be more or less relieved by using data augmentation techniques.
3. Samples with less divergence (images in quite uniform background), as shown in figures in the paper, this actually an advantage for the proposed method, once samples are in complex background, does the method still work well?
4. The figures in Fig. 9b can be more smooth.
5. Regarding Detecting the Objects of Irregular Shapes, there are a lot of CNN based works (e.g., crack detection), should cite and make comparison.
Reviewer 2 Report
A very interesting paper with some problems in explaining. I suggest authors to review this manuscript extensively and make the text more clean & understandable.
Starting with the oversized and lower quality drawings and misplaced captions, then onto the clunky and overly long text . Text should be rewritten in a more concise way. Paper structure must be enhanced greatly to better represent the work.
This work explains a method that makes use of handcrafted pixel neighbourhood features in order to identify pores (of circular shapes with low density) and fractures (of rectangular shapes with low density) in rock formations in the crust of the earth. The authors then went on to test their methods (and a few others) by putting a very big penalty for any data-hungry method: very low quality images and very few samples.
Since I am no expert in such geological phenomena, I think I am more than qualified in image processing and ML aspects of the proposal.
The proposed method does not seem novel in the general signal processing and ML literature, however it may be in the geological computation field, therefore I am not pushing for a rejection solely on novelty.
My suggestion is major revision, given the authors can enhance the paper by following my dire observations:
1- You don't need to make DL comparisons in the current scenario, since you have a very limited data:
The strict restriction of the size of the dataset simply disavows usage of deep-learning based methods for training; therefore if you must compare deep-learning based models, you can only test by trying and using already developed models with some tweaks in data preparations.
If there are no ready to deploy models, which I assume is the case since you trained them on the very limited amount of data you have, the manuscript should be written without any comparison to dl based methods. If you wish to necessarily declare their inefficiency, you can specifically make a paragraph about your opinions, but I'd suggest you make them very clear and brief.
2- You don't need to discredit other methods "because high quality data may not be available". You can simply mention that their effectiveness under such (low-q grayscale) scenarios is not currently known and you can't do analysis on the subject because, "there are no shared CNN models and/or big datasets to train and test them".
3- Scientifically, your proposal method does and can work as expected because it is a hand-crafted and expert-assisted method. In main ML based challenges, the method with which suits the data better, handcrafted by the expert can work better than any other generic and less-tuned method in practice. In ML the challenge is presenting a way to auto-tune as many aspects as possible. Since you mention that some thresholds may be chosen by examiners, this may well be the case of the data you have with the method you propose. This is not to say the proposed method would work similarly-well or much worse in case of a new dataset.
4- A specific comparison on the dataset on which the currently available comparable methods in the related literature were tested against yours can be useful. You are testing on grayscale CT-scans of rocks only, what did others do? Color camera images? Are they using compressed images (like jpegs) or raw data (like a csv or any other uncompressed version of data)?
5- Can you name, if available, any such public datasets regarding rock textures (pores and fractures) in the related literature? If so, why didn't you use them? If there is none, can you make a sentence that there is none, and explain why that would be the case?
6- A proper comparison was still possible, but I'm understanding you didn't explore that venue: You can scriptise an algorithm to make use of your data and multiply it through various data augmentation methods.
The critical failure of the paper is not addressing the issue of data availability more clearly. Why should having few and very low-quality samples be a necessity? If so, why can't data be augmented? Or what would your preferred sample size be?
Round 2
Reviewer 1 Report
The authors have properly answered most my concerns, I have no more questions. Suggest to accept the paper after text editing.
Author Response
Authors response: The authors are thankful to the reviewer for his/her valuable suggestions and confirmation that we were able to address your feedback/comments. Further we are undergoing English revision in parallel and expected to receive the feedback from English review company on or before 27th December 2022 soon after we will update the final manuscript.
Reviewer 2 Report
The authors are proposing a way to identify fractures and pores of rock formations.
The study area is interesting.
The method is straightforward.
Apart from issues with incomplete sentences throughout the paper, I'd like to stress the following matters for authors to think about:
Let me first start with Algorithm 1.
In this algorithm, the input is "Grayscale image of rock contain l × s pixels".
However, the image is then turned into grayscale in the loop.
Did image change it's course within the algorithm?
With the context from the text, the images are generally already grayscale therefore, \forall R=G=B. but the algorithm was designed to handle RGB color images even though it states it's for Grayscale images. (1)
And, in the same algo, the "I" variable (capital i) is not defined. Is it a threshold? The algorithm doesn't take it as an input nor as a parameter; is it a temporary variable generated within? The algorithm doesn't generate it within. (2)
Same applies to Algorithm 2: Are they color images? If so, why a rgb conversion applied here? Since the input image is supposed to be grayscale, then you are basically multiplying grayscale inputs with 1.03, %3 higher than their values. Is this a factor?
Please reconsider the abovementioned issues. These are very obvious errors with easy resolutions.
Author Response
Comments and Suggestions for Authors
The authors are proposing a way to identify fractures and pores of rock formations. The study area is interesting. The method is straightforward.
Apart from issues with incomplete sentences throughout the paper, I'd like to stress the following matters for authors to think about:
- Comment 1:
- Let me first start with Algorithm 1. In this algorithm, the input is "Grayscale image of rock contain l × s pixels". However, the image is then turned into grayscale in the loop. Did image change it's course within the algorithm? With the context from the text, the images are generally already grayscale therefore, \forall R=G=B. but the algorithm was designed to handle RGB color images even though it states it's for Grayscale images. (1)
- Authors response: The authors are thankful to the reviewer for his/her valuable suggestions and highlighting this point. Images were originally grayscale and RGB reference was added to show how color RGB images can be converted to Grayscale. We agree that RGB reference is incorrect hence making the change by removing it from the algorithm and from the text referring to it for example deleted = × 0.299 + × 0.587 + × 0.114 and any reference to RGB in the paper.
- Comment 2:
- And, in the same algo, the "I" variable (capital i) is not defined. Is it a threshold? The algorithm doesn't take it as an input nor as a parameter; is it a temporary variable generated within? The algorithm doesn't generate it within. (2)
- Authors response: The authors are thankful to the reviewer for his/her valuable suggestions. Yes, “I” is a threshold use to binarize the image and as mentioned in the Section II> Classifying Bright and Dark Pixels. We proposed readers to choose suitable “I” value based on their expert judgments and result validation as a parameter.
- Comment 3: The authors are thankful to the reviewer for his/her valuable suggestions.
- Same applies to Algorithm 2: Are they color images? If so, why a rgb conversion applied here? Since the input image is supposed to be grayscale, then you are basically multiplying grayscale inputs with 1.03, %3 higher than their values. Is this a factor?
- Authors response: On the RGB comment pls refer to our response for Comment 1. Referring to multiplication with 1.03, these are the hyperparameter of the algorithm which are to be decided to optimize the model performance against the base metrics like compactness, curl, width and area. The value like 1.03, is arrived at by exploring the model response during validation process. These metrics are further combined in various ways (as highlighted in Algorithm2) to classify a bag of pixel (Hi) as a pore or a fracture.
- Overall Comment: Please reconsider the above mentioned issues. These are very obvious errors with easy resolutions.
- Authors response: We tried our best to address your comments hope we are able to meet your expectations. Further we are undergoing English revision and expected to receive the feedback from English review company on or before 27th Dec 2022. Pls let us know if you have any more feedback or guidance.
Further we are adding tracked manuscript for your reference highlighting changes.
